# Cortical dynamics in hand/forelimb S1 and M1 evoked by brief photostimulation of the mouse's hand

**Daniela Piña Novo[1]\*, Mang Gao[1], Rita Fischer[1], Louis Richevaux[1], Jianing Yu[2], John M Barrett[1][†], Gordon MG Shepherd[1]\*[†]**

[1]Department of Neuroscience, Feinberg School of Medicine, Northwestern University, Chicago, United States; [2]School of Life Sciences, Peking University, Beijing, China

**\*For correspondence:**
danielanovo@northwestern.edu (DPN);
g-shepherd@northwestern.edu (GMGS)

[†]These authors contributed equally to this work

**Competing interest:** The authors declare that no competing interests exist.

## eLife Assessment

This work defines the response dynamics in forepaw-related cortical circuits of S1 and M1 following stimulation of peripheral mechanoreceptors in the mouse. In this revised version, the authors have addressed the reviewers' prior concerns. The results are **convincing** and present a **valuable** comparison to previously published work. This study has implications for understanding the inter-actions between primary somatosensory and motor cortex, required for active sensing, and will be of interest to scientists seeking to better understand the functions of somatosensory and motor circuits.

**Abstract** Spiking activity along synaptic circuits linking primary somatosensory (S1) and motor (M1) areas is fundamental for sensorimotor integration in cortex. Circuits along the ascending somatosensory pathway through mouse hand/forelimb S1 and M1 were recently described in detail (Yamawaki et al., 2021). Here, we characterize the peripherally evoked spiking dynamics in these two cortical areas. Brief (5 ms) optogenetic photostimulation of the hand generated short (~25 ms) barrages of activity first in S1 (onset latency 15 ms) then M1 (10 ms later). The estimated propagation speed was 20-fold faster from hand to S1 than from S1 to M1. Amplitudes in M1 were strongly attenuated. Responses were typically triphasic, with suppression and rebound following the initial peak. Evoked activity in S1 was biased to middle layers, consistent with thalamocortical connectivity, while that in M1 was biased to upper layers, consistent with corticocortical connectivity. Parvalbumin (PV) inhibitory interneurons were involved in each phase, accounting for three quarters of the initial spikes generated in S1, and their selective photostimulation sufficed to evoke suppression and rebound in both S1 and M1. Partial silencing of S1 by PV activation during hand stimulation reduced the M1 sensory responses. Overall, these results characterize how evoked spiking activity propa-gates along the hand/forelimb transcortical loop, and illuminate how in vivo cortical dynamics relate to the underlying synaptic circuit organization in this system.

## Introduction

Somatosensory signals reach primary motor cortex (*Goldring et al., 1970*; *Strick and Preston, 1982*; *Swadlow, 1994*; *Naito et al., 2002*; *Ferezou et al., 2007*; *Hatsopoulos and Suminski, 2011*; *Mao et al., 2011*; *Murray and Keller, 2011*; *Alonso et al., 2023*; *Reddy et al., 2024*). To do so, signals from mechanoreceptor afferents ascend primarily via lemniscal pathways and somatosensory thalamus to arrive in primary somatosensory cortex (S1), which sends corticocortical projections to primary motor

cortex (M1). The thalamus→S1→M1 circuits are well studied in model systems such as the rodent vibrissal pathways (*Ferezou et al., 2007*; *Mao et al., 2011*; *Sreenivasan et al., 2016*; *Petersen, 2019*). We recently confirmed and extended these findings for the corresponding circuits of mouse hand/forelimb S1 and M1, using slice-based optogenetic and electrophysiological methods to delineate synaptic circuit connections along the transcortical sensorimotor loop (*Yamawaki et al., 2021*).

A salient feature of the hand/forelimb circuits is the transition, on reaching S1 cortex, from a 'streamlined' organization of the ascending subcortical circuits via thalamus to a 'densely polysynaptic' organization of the local and interareal cortical circuits (*Yamawaki et al., 2021*). Here, we sought to characterize the timing, amplitude, and related properties of the cortical dynamics in the same S1 and M1 circuits, evoked by stimulating hand mechanoreceptors. We developed an optogenetic–electrophysiological approach guided by the desiderata of delivering brief stimuli to awake animals during simultaneous recordings of spiking activity in both areas, leveraging a mouse line enabling cortical as well as peripheral optogenetic manipulations. This approach enabled quantitative characterization of fundamental properties of the spiking dynamics of hand/forelimb S1 and M1 in response to brief peripheral stimulation.

## Results

### Brief stimulation of the hand evokes barrages of spiking activity in S1 and M1

To study cortical responses to hand stimulation, we aimed to develop a paradigm for delivering short-duration stimuli while electrophysiologically sampling spiking activity in hand/forelimb S1 and M1 cortical areas of awake mice. The rationale for using brief stimuli was to obtain the equivalent of an impulse response function of the system. For example, with millisecond-scale stimulation, essentially all activity, including any longer-lasting responses extending in time well after the initial pulse of input, can be ascribed to the initial input alone. We adopted an optogenetic approach for delivering 'phototactile' somatosensory stimuli to the mouse's hand with millisecond precision, combining this with cortical electrophysiological recordings of evoked spiking activity (**Methods**). Motivations for this

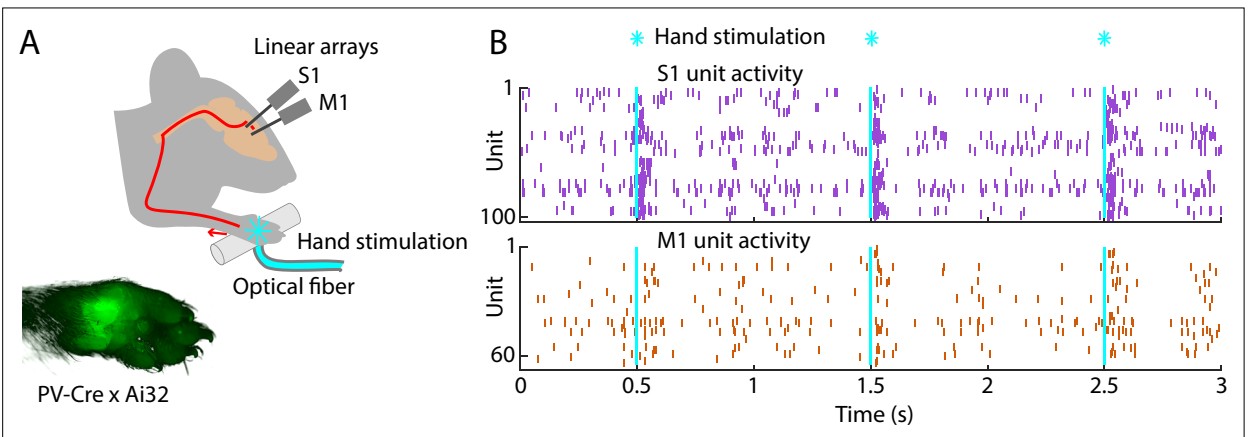

**Figure 1.** Brief stimulation of the hand evokes barrages of spiking activity in S1 and M1. (**A**) Experimental approach. Schematic depicts optogenetic photostimulation of the hand of an awake head-fixed mouse via an optical fiber in the holding bar, with cortical population spiking activity recorded on linear arrays inserted in both S1 and M1. Image of the hand shows green fluorescence across the glabrous skin of the palm, from labeling of mechanoreceptor afferents (PV-Cre × Ai32 mouse). (**B**) Example segment of a recording, showing for three trials the raster plots of spiking activity for active units in S1 (top, purple) and M1 (bottom, orange) during photostimulation of the hand (25 trials total, 1 s inter-stimulus interval, 5 ms duration, 5 mW light intensity at the fiber tip, 910 μm core diameter).

The online version of this article includes the following video and figure supplement(s) for figure 1:

**Figure supplement 1.** Optogenetic stimulation of mechanosensory afferents in the mouse's hand does not evoke forelimb movements.

**Figure supplement 2.** Histological reconstruction of electrode placements in S1 and M1.

**Figure 1—video 1.** Video motion analysis.

https://elifesciences.org/articles/105112/figures#fig1video1

approach included the ease of precisely controlling stimulus timing and intensity, and compatibility with awake animals.

Crossing a proprioceptor-specific driver line of mice (PV-Cre) with a channelrhodopsin-2 (ChR2) reporter line (Ai32) yielded expression of ChR2 in peripheral mechanoreceptor afferents, particularly around the palm (*Figure 1A*, **Methods**). We chose this line because it fortuitously also labels parvalbumin-type GABAergic neurons in the cortex, enabling cortical optogenetic manipulations as well. To sample stimulus-evoked cortical responses, we recorded from multi-channel linear arrays inserted in S1 and M1 of awake, head-fixed mice as they rested their hands on holding bars harboring an optical fiber through which light stimuli were delivered to the palm. As shown in an example recording, photostimulation of the hand with 5-ms pulses of blue light elicited barrages of cortical spiking activity in both areas (*Figure 1B*). Although hand stimulation evoked activity in forelimb M1, it did not evoke movements of the forelimb (*Figure 1—video 1*; *Figure 1—figure supplement 1*). Following experiments, brains were sectioned and inspected to stereotaxically localize the fluorescent tracks of the dye-coated linear arrays to confirm the recording locations (*Figure 1—figure supplement 2*). This approach thus provides a way to record and analyze S1 and M1 cortical activity evoked by brief photostimulation of the hand in awake mice.

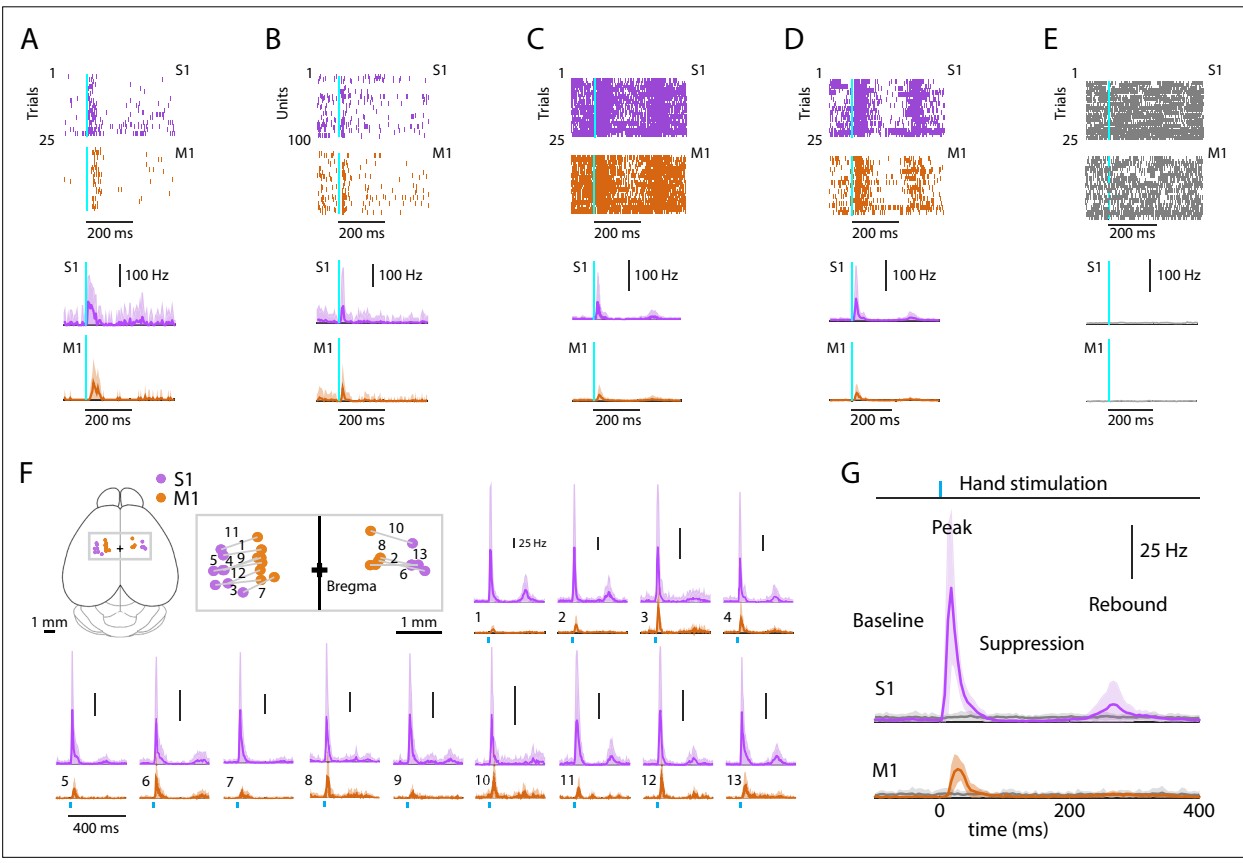

**Figure 2.** Evoked cortical responses follow a triphasic peak-suppression-rebound pattern. (**A**) Top: Example rasters of one stimulus-responsive unit in each area, ordered by trial and aligned to the onset of photostimulation. Bottom: Peristimulus time histogram (PSTH) of average firing rate across trials (mean ± s.d.). (**B**) Top: All stimulus-responsive units on the probes for one trial. Bottom: Average PSTH across units (mean ± s.d.). (**C**) Top: All units on the probes, for all trials. Bottom: Overall average PSTH (mean ± s.d.). (**D**) Same as in C, but only including stimulus-responsive units. (**E**) Top: All units on the probes, for all trials with the hand off the light-delivery bar. Bottom: Overall average PSTH (mean ± s.d.). (**F**) PSTHs for each pairwise S1–M1 recording. Inset (top left) shows brain schematic with locations of the S1 and M1 probes for each experiment. Bottom: Average (mean ± s.d.) PSTHs of responsive units aligned to the photostimulation (dashed line), for each experiment (13 recordings from 9 mice). (**G**) Average stimulus-evoked cortical responses in forelimb S1 and M1. Grand average (mean ± s.d.) PSTHs across recordings aligned to the onset of the hand photostimulation. Light gray traces: same, but for trials with the hand off the light-delivery bar.

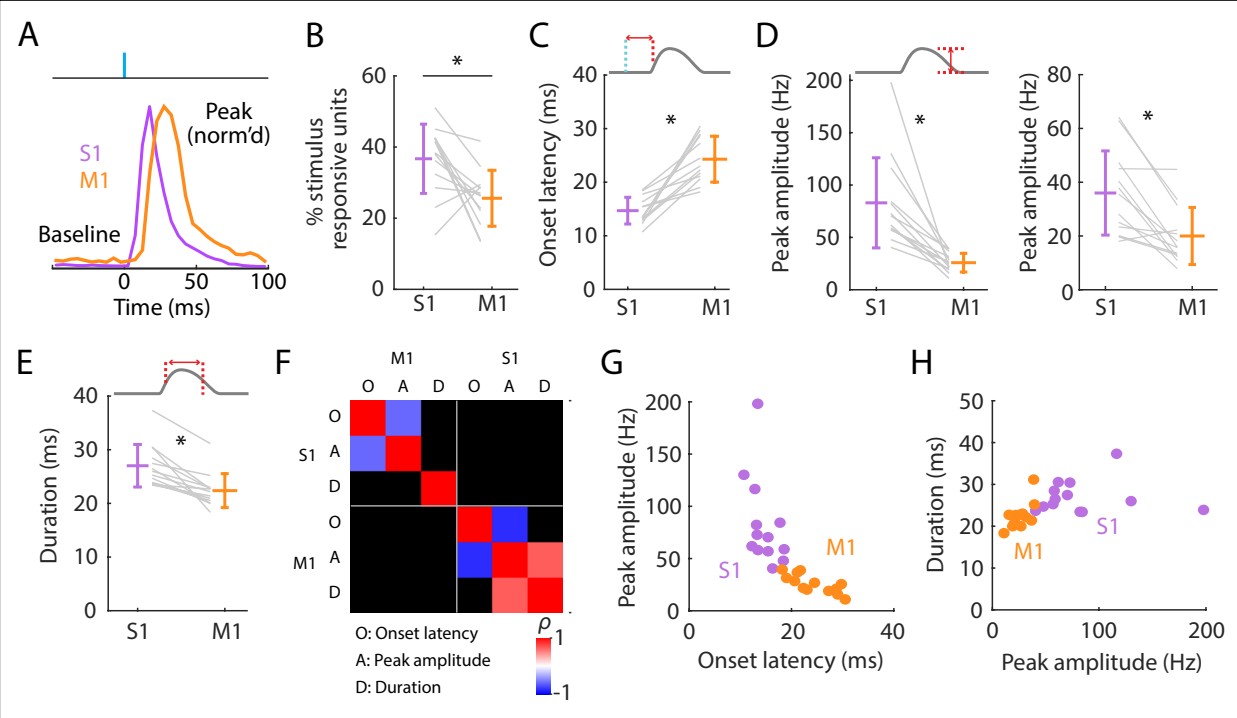

**Figure 3.** Initial peak responses in M1 are delayed and attenuated relative to S1. (**A**) Scaled versions of the grand average peristimulus time histograms (PSTHs), showing the first 100 ms of the responses. (**B**) Percentages of all recorded units that were stimulus responsive, with the overall averages (mean ± s.d.) and group comparison (Wilcoxon's signed rank test, $W = 80$, *: $p = 0.01$; $n = 13$ recordings from 9 mice). (**C**) Onset latencies in S1 and M1, averaged over individual responsive units for each experiment (gray lines), along with the overall averages (mean ± s.d.) and group comparison ($W = 0$, *: $p = 2 \times 10^{-4}$). The schematic above the plot depicts measurement of onset latency relative to stimulus onset. (**D**) Left: Same as C, but for peak amplitudes ($W = 91$, *: $p = 2 \times 10^{-4}$). Right: Same, but calculated for single units only ($W = 88$, *: $p = 0.01$). (**E**) Same as C, but for response durations ($W = 91$, *: $p = 2 \times 10^{-4}$). (**F**) Correlation matrix showing correlations among average response properties for all recordings (mean values across individual responsive units for each experiment). Colors indicate Spearman's correlation index, rho ($\rho$). Non-significant correlations are shown in black. (**G**) Peak amplitude versus onset latency (S1: $\rho = -0.58$, $p = 0.04$; M1: $\rho = -0.82$, $p = 0.001$). (**H**) Response duration versus peak amplitude (S1: $\rho = 0.02$, $p = 0.95$; M1: $\rho = 0.64$, $p = 0.02$).

## Evoked cortical responses follow a triphasic peak-suppression-rebound pattern

To characterize the cortical responses, we recorded from hand/forelimb S1 and M1 in a cohort of animals and analyzed the evoked activity; except where otherwise indicated, results reported below are all based on this sample (13 recordings from 9 animals), and analyses focused on stimulus-responsive units, with pooling of multi- and single units as 'active units' (**Methods**) (*Barrett et al., 2022*). Responses to hand photostimulation in multiple (generally 25) trials were analyzed based on alignment to the onset of photostimulation, to generate average peristimulus time histograms (PSTHs) for each responsive unit on each probe, as shown for one example recording (*Figure 2A–E*), for each experiment as average PSTHs (*Figure 2F*), and as grand average PSTHs across all recordings (*Figure 2G*). In trials where the mouse's hand was not resting on the light-delivery bar, no responses were observed (*Figure 2E, G*). This also served as a control for potential visual responses to the flashing light. As seen in the individual experiments as well as in the grand averages, peripheral stimulation evoked a triphasic response in both areas, with an initial short-latency large-amplitude peak followed by an interval of suppressed activity and a subsequent late lower-amplitude rebound. We proceeded to analyze the multiple components of the evoked responses to hand stimulation in greater detail.

## Initial peak responses in M1 are delayed and attenuated relative to S1

Focusing first on the short-latency initial peak, we calculated basic response parameters for individual responsive active units for each experiment, and as grand averages across experiments (*Figure 3*;

**Table 1.** Properties of the evoked S1 and M1 responses.

| Parameter | S1 | M1 | S1 vs M1 statistical comparison | M1 minus S1 difference |
|---|---|---|---|---|
| Baseline firing rate (Hz) | 1.2 ± 0.7 | 0.6 ± 0.2 | 0.001 | −0.6 ± 0.7 |
| *Short-latency response* | | | | |
| Stimulus-responsive units (%) | 36.7 ± 9.7 | 25.6 ± 7.9 | 0.01 | n/a |
| Onset latency (ms) | 14.7 ± 2.5 | 24.3 ± 4.3 | $2 \times 10^{-4}$ | 9.6 ± 5.3 |
| Peak latency (ms) | 21.4 ± 2.5 | 31.2 ± 4.3 | $2 \times 10^{-4}$ | 9.8 ± 5.1 |
| Duration (ms) | 27.0 ± 4.0 | 22.4 ± 3.2 | $2 \times 10^{-4}$ | −4.6 ± 3.0 |
| Amplitude (Hz) | 83.0 ± 43.1 | 25.8 ± 8.9 | $2 \times 10^{-4}$ | n/a |
| *Post-peak suppression* | | | | |
| Stimulus-responsive units with suppression (%) | 22.3 ± 13.6 | 8.4 ± 7.7 | 0.01 | n/a |
| Amplitude (Hz) | 0.3 ± 0.3 | 0.1 ± 0.2 | 0.2 | n/a |
| Amplitude (% of baseline) | 8.2 ± 6.3 | 5.6 ± 7.5 | 0.4 | n/a |
| *Post-inhibitory rebound* | | | | |
| Stimulus-responsive units with rebound (%) | 30.9 ± 22.1 | 8.1 ± 9.6 | $2 \times 10^{-4}$ | n/a |
| Peak latency (ms) | 289.0 ± 31.9 | 274.7 ± 36.2 | 0.8 | −2.4 ± 21.9 |
| Amplitude (Hz) | 31.4 ± 13.6 | 20.2 ± 5.3 | 0.02 | n/a |

Parameters of the activity in S1 and M1 evoked by hand photostimulation. Measurements are reported as the overall average (mean ± s.d.) across recordings (13 recordings from 9 mice) along with the group comparison (Wilcoxon's signed rank test).

*Table 1*). Prior to stimulus onset, baseline firing rates were low in S1 and even lower in M1. Following stimulus onset, S1 activity rapidly peaked, followed by M1 activity (*Figure 3A*). Only a subset of units was stimulus responsive in both areas, on average more in S1 (36.7%) than in M1 (25.6%) (*Figure 3B*). Responses differed substantially in onset latency (14.7 ms in S1, 24.3 ms in M1) (*Figure 3C*). Smaller peak amplitudes were found in M1 (for all responsive units: 25.8 Hz; responsive single units only: 20.0 Hz) compared to S1 (for all responsive units: 83.0 Hz; responsive single units only: 36.0 Hz) (*Figure 3D*). The average ratio of M1 over S1 peak amplitude was roughly half (for all responsive units: 0.36 ± 0.16, mean ± s.d.; responsive single units only: 0.60 ± 0.29). Response durations were brief and slightly longer in S1 (27.0 ms) than in M1 (22.4 ms) (*Figure 3E*). These patterns were consistent across experiments.

We explored which response features may be correlated (*Figure 3F*). Perhaps unsurprisingly, both in S1 and M1, peak amplitude strongly correlated with onset latency (*Figure 3G*) and peak latency (S1: $\rho = -0.69$, p = 0.01; M1: $\rho = -0.71$, p = 0.01); the faster the onset, the larger the amplitude. Few other significant correlations were found, however. Response durations were not significantly correlated with peak amplitudes in S1, where responses were generally strongest, and only relatively weakly correlated in M1 (*Figure 3H*). Thus, larger responses terminated just as quickly as smaller ones. Although response amplitudes were consistently bigger in S1 than in M1 they were not significantly correlated with each other. This may reflect variability in probe placements, the numbers and types of units recorded, and other factors contributing to animal-to-animal variability.

Having measured the latencies of the evoked spiking activity reaching S1 and M1, we also estimated the corresponding effective propagation speeds. To do so, we estimated the hand-to-S1 and S1-to-M1 pathway distances and divided these by the latencies (**Methods**). The hand-to-S1 distance was estimated as 44.3 ± 1.1 mm, and the hand-to-S1 propagation speed was accordingly 3.0 ± 0.1 m/s based on onset latencies (2.1 ± 0.1 m/s based on peak latencies). The S1-to-M1 distance, measured as the Euclidean distance between the probes, was 0.92 ± 0.12 mm (mean ± s.d.), and the estimated S1-to-M1 propagation speed was accordingly 0.14 ± 0.11 m/s based on the difference (M1 minus S1) in onset latencies (0.14 ± 0.10 m/s based on peak latencies). Thus, the propagation speed for the entire subcortical leg of the ascending pathway was approximately 20-fold faster than for the short corticocortical leg, from S1 to M1.

These results quantify basic properties of the initial component of the evoked cortical spiking activity in S1 and M1. In S1, responses begin ~15 ms after stimulus onset and reach peak amplitude ~6 ms later. In M1, responses have a similar temporal profile but with a ~10 ms lag and approximately one-third of the S1 amplitude. Thus, M1 receives a slightly delayed and substantially attenuated short-latency somatosensory response from brief hand stimulation. In both areas, responses self-terminate, extinguishing within ~25 ms.

## Post-peak activity is suppressed in S1 and rebounds in both S1 and M1

Focusing next on the later phases of the evoked responses, we noted that the initial burst of evoked activity was self-terminating, and particularly in S1 often then fell below the pre-stimulus baseline level as well (*Figure 4A*). The fraction of all stimulus-responsive units that showed post-peak suppression was relatively small (22.3% in S1, 8.4% in M1), but activity of these units was greatly suppressed (to 8.2% of baseline activity level in S1, and 5.6% in M1) (*Figure 4B, C*; *Table 1*). In both areas, activity also tended to rebound following suppression, more prominently in S1 than in M1 and approximately simultaneously (*Figure 4D–F*; *Table 1*). Rebound activity in S1 was observed in 30.9% of stimulus-responsive units, firing on average at 31.4 Hz, while rebound activity in M1 involved 8.1% of stimulus-responsive units firing at 20.2 Hz (rebound firing rates based on single units only could not be calculated due to low sample size).

We explored correlations and scaling relationships among the components of the triphasic responses (*Figure 4G*). Rebound amplitude and initial peak amplitude were strongly correlated in both S1 and M1 (*Figure 4H*). In S1, the proportion of units showing suppression correlated with the amplitude of the initial peak (*Figure 4I*) but not with the rebound (*Figure 4J*). Rebound timing (latency) was not strongly correlated with any parameters, except for rebound amplitude in M1 only (*Figure 4G*). Other correlations were weaker.

This triphasic peak-suppression-rebound pattern implies that after the initial sensory response the cortex becomes inhibited, and then disinhibited as inhibition self-terminates, as observed in other studies of sensory cortex (see Discussion). To explore this, we tested the sensitivity of the cortex to a second photostimulus, timed to coincide with the intervals of either the suppressed or rebound activity, or later (i.e., a paired-pulse paradigm, with variable lag) (*Figure 4K–N*). In both S1 and M1, responses to the second stimulus fell to approximately half the amplitude of the first during the suppressed interval, returning to normal during the rebound interval and after (*Figure 4M, N*).

These results show that both S1 and M1 excitability recovers by 250 ms after hand photostimulation. This implies that somatosensory input generates in S1, and to a lesser extent in M1, a sequence of excitation, inhibition, and disinhibition, with the two later stages scaling in proportion to the initial response amplitude. The self-terminating property of the initial large-amplitude response together with the ensuing suppression of activity suggests strong initial recruitment of inhibitory interneurons. Therefore, we next investigated the response properties of cortical interneurons, using cell-type-specific methods to identify them in the recordings and manipulate their activity.

## Laminar profiles of evoked activity in S1 and M1

To assess the laminar distribution of the evoked activity, we used the same methods as in the probe-averaged analyses described above but with spiking responses of active units binned according to cortical depth (11 recordings). In S1, evoked activity was distributed somewhat broadly across cortical layers, with the strongest activity biased to the middle layers (*Figure 5A*). In M1, evoked activity was also distributed somewhat broadly across cortical depth, with a bias to middle and upper layers, particularly layer 2/3 (*Figure 5B*). In both areas, the laminar profile of the rebound responses resembled, and were strongly correlated with, those of the initial peak responses (*Figure 5—figure supplement 1*).

## PV neurons in S1 are strongly recruited by hand stimulation

In the same recordings, in nearly every experiment (12 recordings from 8 mice) we also used opto-tagging to identify a subset of inhibitory neurons within the population, focusing on recordings in S1, where the suppression effects were strongest, as described above. Conveniently, in the same PV-Cre × Ai32 mice in which peripheral mechanoreceptor afferents are labeled, cortical parvalbumin-expressing (PV) interneurons are also labeled (*Figure 6—figure supplement 1*). Thus, application of

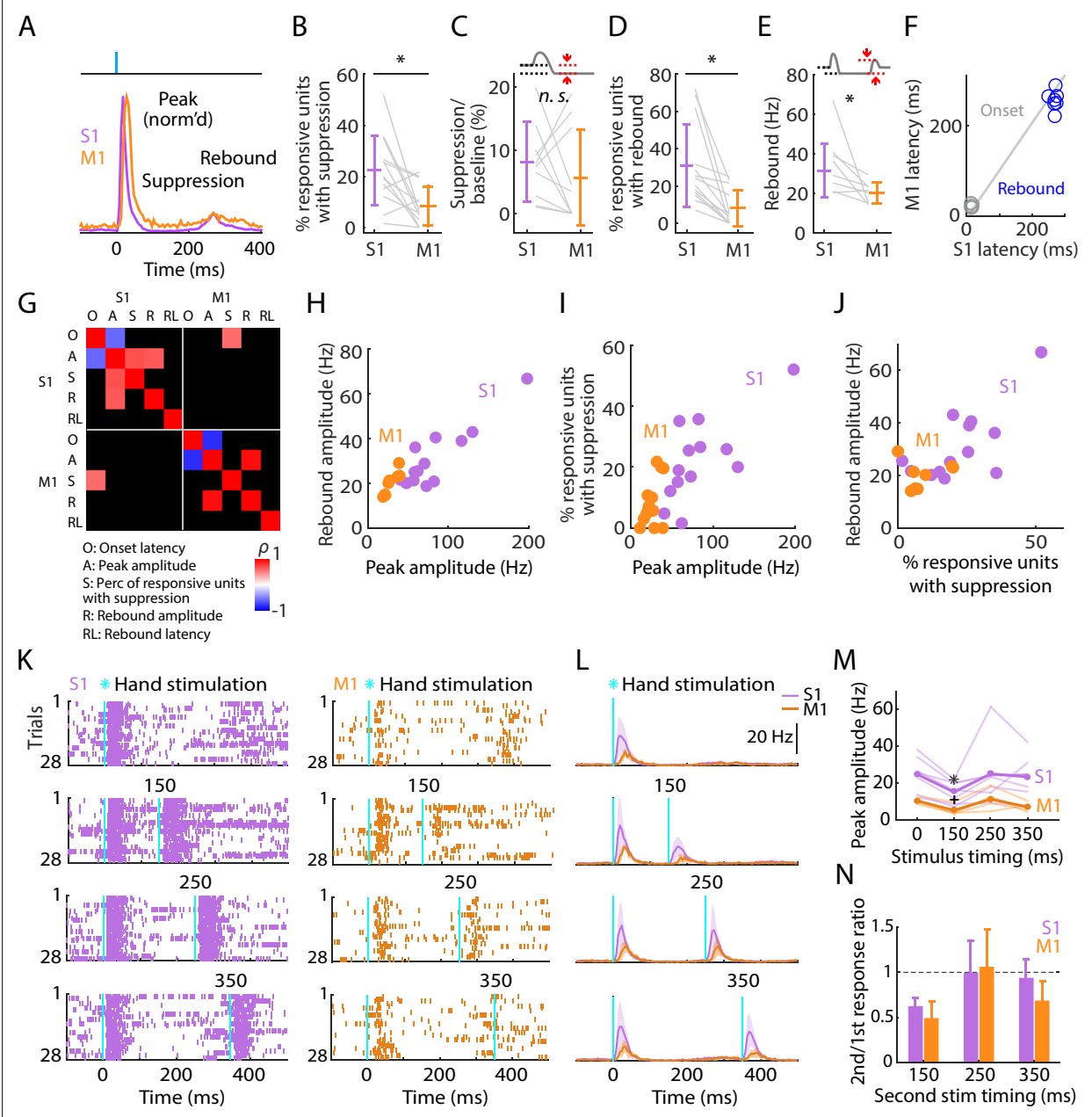

**Figure 4.** Post-peak activity is suppressed in S1 and rebounds in both S1 and M1. (**A**) Scaled versions of the grand average peristimulus time histograms (PSTHs), showing the first 400 ms of the responses. (**B**) Percentage of stimulus-responsive units in S1 and M1 exhibiting significant post-peak suppression (relative to the pre-stimulus baseline), for each experiment (gray lines) along with the overall averages (mean ± s.d.) and group comparison (Wilcoxon's signed rank test, $W = 82$, *: $p = 0.01$; $n = 13$ recordings from 9 mice, stimulus-responsive units only). (**C**) Suppression as a percentage of baseline firing rate amplitude for units in B. The schematic above the plot depicts measurement of amplitude during the suppression period (from 110 to 170 ms post-stimulation) ($W = 36$, $p = 0.43$). (**D**) Percentage of stimulus-responsive units exhibiting significant post-suppression rebound (relative to the pre-stimulus baseline), for each experiment (gray lines) along with the overall averages (mean ± s.d.) and group comparison ($W = 91$, *: $p = 2 \times 10^{-4}$). (**E**) Rebound amplitude for units in D ($W = 35$, *: $p = 0.02$). (**F**) Average onset and rebound latencies in M1 versus S1, for each experiment (circles). (**G**) Correlation matrix showing correlations among average response properties for all recordings (mean values across individual responsive units for each experiment). Colors indicate Spearman's correlation index, rho ($\rho$). Non-significant correlations are shown in black. (**H**) Rebound amplitude versus peak amplitude (S1: $\rho = 0.65$, $p = 0.02$; M1: $\rho = 0.95$, $p = 0.001$). (**I**) Percentage of stimulus-responsive units with suppression versus peak amplitude (S1: $\rho = 0.68$, $p = 0.01$; M1: $\rho = 0.38$, $p = 0.2$). (**J**) Rebound amplitude versus percentage of stimulus-responsive units with suppression (S1: $\rho = 0.52$, $p = 0.07$; M1: $\rho = 0.14$, $p = 0.75$). (**K**) Example segment of a recording, showing raster plots of population spiking activity in S1 and M1, aligned to single (top) and double photostimulation of the hand (28 trials total). (**L**) Grand average (mean ± s.d.) PSTHs across recordings, aligned to the onset of the hand photostimulation ($n = 7$ recordings from 4 mice in S1, and 4 recordings from 3 mice in M1). (**M**) Peak amplitudes in S1 and M1 evoked by a single

*Figure 4 continued on next page*

*Figure 4 continued*

stimulus (at time 0) and a second stimulus delivered with variable lag (150, 250, or 350 ms after the first), for each experiment (thin lines) along with the overall averages (thick lines, mean), and group comparisons (Friedman test, S1 latency effect, $\chi_3^2 = 8.66$, p = 0.03; M1 latency effect, $\chi_3^2 = 9.3$, p = 0.02; Dunn–Sidak's post hoc multiple comparisons within S1, *: p = 0.04, and M1, +: p = 0.04 for second pulse at 150 ms versus first pulse; no other significant post hoc differences were found). (**N**) S1 and M1 peak amplitudes evoked by the second pulse, normalized to the first pulse's response (horizontal dashed line).

blue-light photostimuli to S1 while recording local population spiking activity enabled detection of activated PV neurons based on short-latency sustained firing responses (**Methods**; *Figure 6A, B*). With this approach we parsed the population spiking activity evoked by hand photostimulation into PV versus non-PV units (*Figure 6C*).

Analysis of this dataset showed that peripheral stimulation activates a population comprising a larger proportion of PV (64.4% of all activated units) than non-PV units (35.6%) (*Figure 6D*). The proportion of stimulus-responsive units among the PV units (64.0%) was also higher than among non-PV units (31.1%) (*Figure 6E*). Furthermore, PV units collectively accounted for most of the evoked spikes (76.1%) (*Figure 6F*). Further analysis showed an interesting mix of differences and similarities in key parameters across the three phases of the stimulus-evoked responses. The initial peak responses arrived with ~5 ms shorter latency for PV than for non-PV units (*Figure 6G*), consistent with the initial positive deflection in the difference PSTH (*Figure 6C*, bottom). The PV units also reached ~2-fold higher peak amplitude (*Figure 6H*). During the suppression phase, the percentage of units exhibiting significant suppression was higher for PV (30.4%) and lower for non-PV units (18.0%) (*Figure 6I*), and consistent with the intermediate value (22.3%) observed for all units, as noted above (*Figure 4B*). The activity of these units, both PV and non-PV, was greatly suppressed (to 7.3% of baseline activity level for PV and to 9.2% for non-PV units) (*Figure 6J*). In the subsequent rebound phase, a higher fraction of PV (37.6%) than non-PV (27.3%) units showed rebound activity (*Figure 6K*). No differences were found in rebound amplitudes or latencies (*Figure 6L, M*).

These results show that PV-type interneurons in S1 are recruited strongly and rapidly during the initial cortical response to hand stimulation. Indeed, the evoked firing of PV units even leads that of non-PV units by several milliseconds, consistent with prior observations of vibrissal S1 cortical responses to whisker deflection (*Yu et al., 2016*). Following the suppression phase of the response, in the rebound phase the PV units are again robustly engaged, with a temporal profile similar to the non-PV units. The prominent role of PV neurons in all phases of the responses raises the possibility that

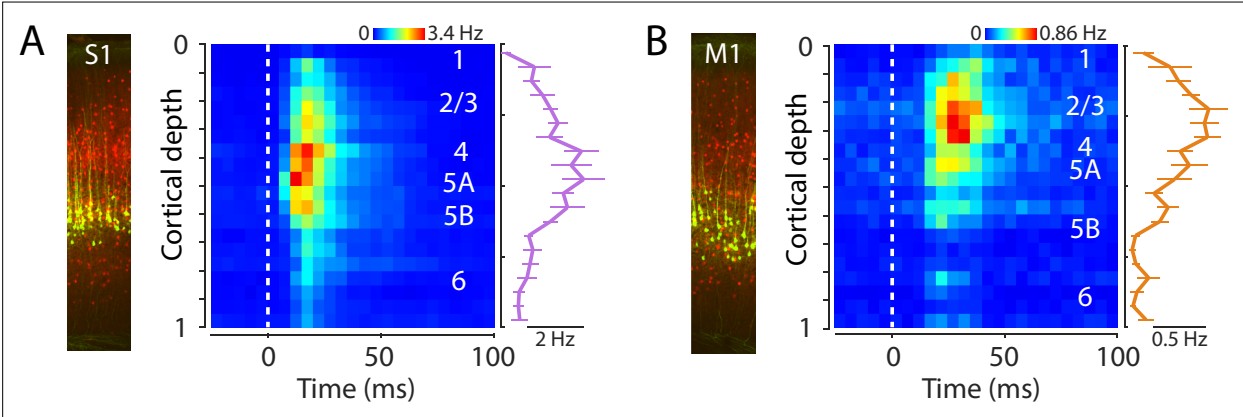

**Figure 5.** Laminar profiles of evoked activity in S1 and M1. (**A**) Left: Example image of S1 cortex with labeled corticospinal neurons (green) and parvalbumin (PV) neurons (red). Right: Evoked spiking activity of all active units across the depth of the cortex (average of 11 recordings from 8 mice). Each unit's spikes were binned according to its depth, in 20 bins total across the full cortical depth, where 0 is the pial surface and 1 is the lower boundary of cortex with white matter. Plot shows the grand average amplitude in each depth bin (mean ± s.e.m. across recordings), measured around the time of the peak response. (**B**) Same, for M1 laminar profile.

The online version of this article includes the following figure supplement(s) for figure 5:

**Figure supplement 1.** Laminar profiles of activity across the triphasic response to hand stimulation.

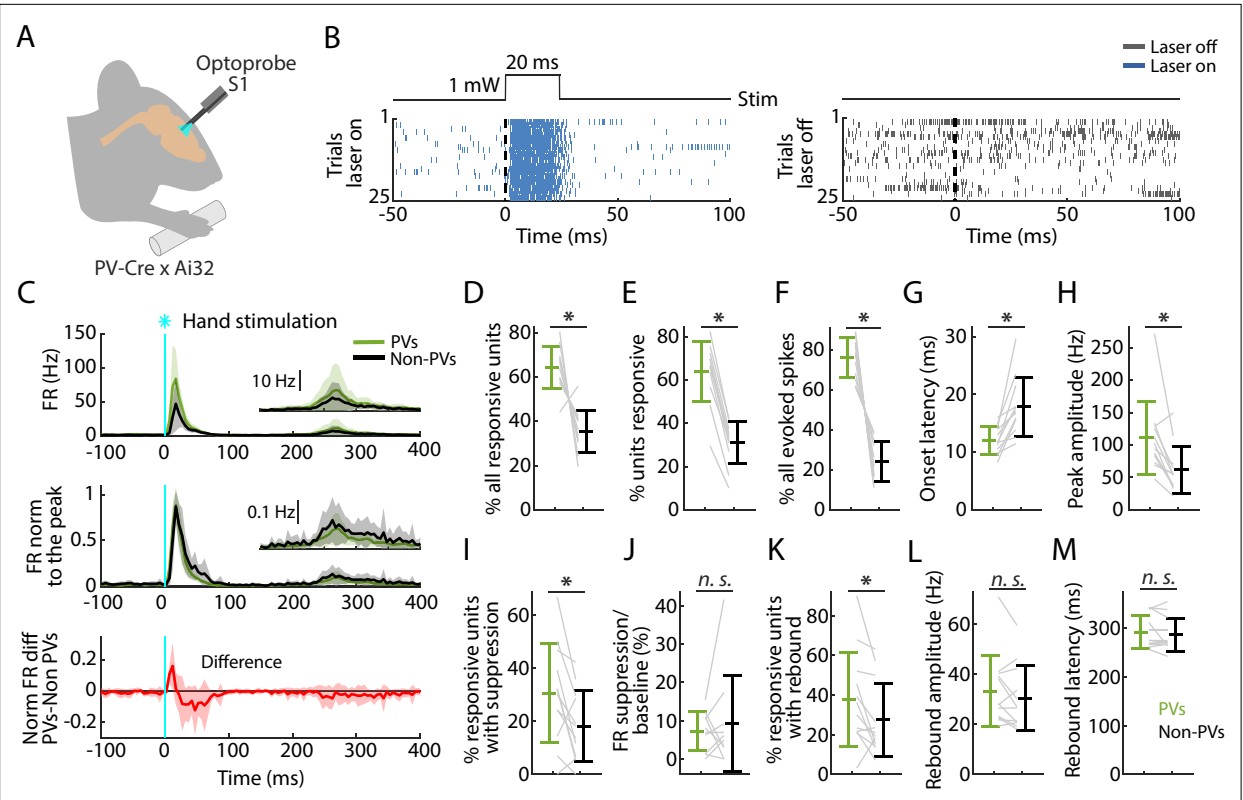

**Figure 6.** PV neurons in S1 are strongly recruited by hand stimulation. (**A**) Experimental approach. Schematic depicts recording during optogenetic stimulation of parvalbumin (PV) neurons in S1. (**B**) Left: Example segment of a recording, shown as a raster plot of population spiking activity during the local optogenetic stimulation (laser on, 25 trials total, 1 s inter-stimulus interval, 20 ms duration, 1 mW light intensity at the fiber tip, 105 µm core diameter). Right: Raster plot of spiking activity aligned to sham events triggered at the same interval as the laser stimulus, but with the laser off. (**C**) Top: Grand average (mean ± s.d.) peristimulus time histograms (PSTHs) for the stimulus-responsive opto-tagged PV (green) and non-PV (black) units in S1 across recordings (12 recordings from 8 mice) aligned to the onset of the hand photostimulation. Inset on the right shows the same data on a magnified y-axis scale. Middle: Grand average (mean ± s.d.) peak-normalized PSTHs. Bottom: Average (mean ± s.d.) difference of PV minus non-PV peak-normalized PSTHs. (**D**) Percentage of stimulus-responsive units that are PV versus non-PV, for each experiment (gray lines) along with the overall averages (mean ± s.d.) and group comparison (Wilcoxon's signed rank test, $W = 77$, *: $p = 0.001$). (**E**) Percentage of PV and non-PV units that are stimulus responsive in each experiment (Wilcoxon's signed rank test, $W = 78$, *: $p = 5 \times 10^{-4}$). (**F**) Percentage of evoked spikes coming from PV and non-PV units over the time-course of the initial peak response ($W = 78$, *: $p = 5 \times 10^{-4}$). (**G**) Onset latencies of PV and non-PV units ($W = 0$, *: $p = 5 \times 10^{-4}$). (**H**) Peak amplitudes of PV and non-PV units ($W = 75$, *: $p = 0.002$). (**I**) Percentage of stimulus-responsive PV and non-PV units with significant suppression (in the time window 110–170 ms) compared to pre-stimulus baseline ($W = 68$, *: $p = 0.02$). (**J**) Suppression as a percentage of baseline firing rate amplitude for PV and non-PV units in I ($W = 23$, $p = 0.7$). (**K**) Percentage of stimulus-responsive PV and non-PV units with significant rebound compared to pre-stimulus baseline ($W = 68$, *: $p = 0.02$). (**L**) Rebound amplitude for PV and non-PV units in K ($W = 55$, $p = 0.2$). (**M**) Rebound latency for PV and non-PV units in K ($W = 40$, $p = 0.6$).

The online version of this article includes the following figure supplement(s) for figure 6:

**Figure supplement 1.** Labeling patterns in PV-Cre mice.

activation of PV neurons alone might emulate some or all aspects of the responses evoked by hand stimulation, which we next explored.

## Selective activation of PV neurons generates suppression and rebound

Using the same mice and cortical photostimulation methods that were used in the opto-tagging experiments described above, we analyzed the effect of focally activating PV units in S1 cortex (12 recordings from 8 mice). As above, we used opto-tagging to parse PV versus non-PV units in S1, and examined their activity along with that of all units recorded simultaneously in M1 (*Figure 7A*). Following the selective activation of PV units, the fraction of S1 units exhibiting significant post-peak suppression was similar for PV (18.1%) and non-PV (19.6%) units (*Figure 7B*). During this interval, the firing rate diminished more for PV (to 7.3% of baseline) than for non-PV (to 10.9%) units (*Figure 7C*). In

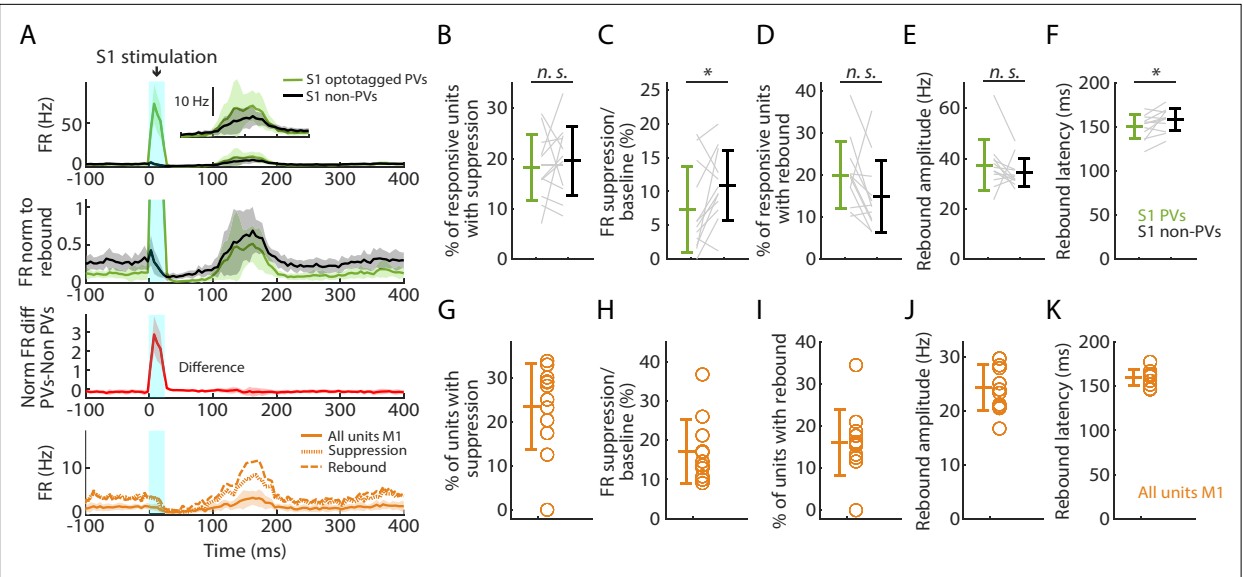

**Figure 7.** Selective activation of parvalbumin (PV) neurons in S1 generates suppression and rebound. (**A**) Top plot: Grand average (mean ± s.d.) peristimulus time histograms (PSTHs) for opto-tagged PV and non-PV units in S1 across recordings (12 recordings from 8 mice), aligned to the onset of the cortical photostimulation (1 mW at the fiber tip, 105 μm core diameter). Inset on the right shows the rebound segment of the same data on a magnified *y*-axis scale. Second plot: Grand average PSTHs normalized to the post peak rebound amplitude. Third plot: Average difference of PV minus non-PV post peak rebound-normalized PSTHs. Fourth plot: Grand average PSTH for all M1 units, and for units with suppression or rebound. (**B**) Percentage of PV and non-PV units with significant suppression (in the time window 110–170 ms) compared to pre-stimulus baseline (Wilcoxon's signed rank test, *W* = 27, p = 0.63). (**C**) Suppression as a percentage of baseline firing rate amplitude for PV and non-PV units in B (*W* = 12, *: p = 0.03). (**D**) Percentage of PV and non-PV units with significant rebound compared to pre-stimulus baseline (*W* = 61, p = 0.09). (**E**) Rebound amplitude for PV and non-PV units in D (*W* = 59, p = 0.13). (**F**) Rebound latency for PV and non-PV units in D (*W* = 10, *: p = 0.02). (**G–K**) Same as B–F but for all units recorded on the M1 probe during S1 photostimulation.

The online version of this article includes the following figure supplement(s) for figure 7:

**Figure supplement 1.** Selective activation of parvalbumin (PV) neurons in M1 generates suppression and rebound.

**Figure supplement 2.** Suppression and rebound induced by parvalbumin (PV) activation scale with photostimulation intensity.

the subsequent phase, 20.0% of PV and 14.9% of non-PV units exhibited rebound activity (*Figure 7D*); no differences were found in rebound amplitudes and minor differences were found in the latencies (*Figure 7E, F*). In M1, the overall activity also followed a pattern of suppression and rebound (*Figure 7A, G–K*).

Similarly, in a subset of experiments (4 recordings from 3 mice), we selectively activated PV neurons in M1. This was also able to induce suppression and rebound locally (in M1), involving both PV and non-PV units (*Figure 7—figure supplement 1*). In both areas, focal stimulation at progressively lower stimulus intensities resulted in reduced responses, and at 25% stimulus intensity the main effect was local activation of PV units in the stimulated area, with little suppression and rebound locally and none in the other (non-stimulated) area (*Figure 7—figure supplement 2*).

These results show that the selective activation of PV neurons in both S1 and M1 cortex can generate a pattern of suppression and rebound closely resembling that observed with hand stimulation.

## Partial silencing of S1 reduces M1 responses to hand stimulation

If M1 responses to hand stimulation depend on S1, then manipulations that reduce S1 activity during hand stimulation should also cause a reduction in the M1 responses. To test this, we performed focal stimulation of S1 PV neurons as above, in this case pairing it with simultaneous hand stimulation (*Figure 8*). As before, hand stimulation alone evoked strong responses in S1 and weaker and delayed responses in M1 (*Figure 8A*). We used low-intensity laser stimulation of the cortex (at 25% of the intensity used for the opto-tagging and cortical silencing experiments above) to isolate the effects of locally activating PV neurons in S1 without suppressing ongoing activity in M1 (*Figure 8B*). When hand stimulation was paired with S1 silencing, the activity in S1 followed a hybrid pattern (reflecting the combined effect of activating the PV units and partially suppressing the sensory-evoked response), and

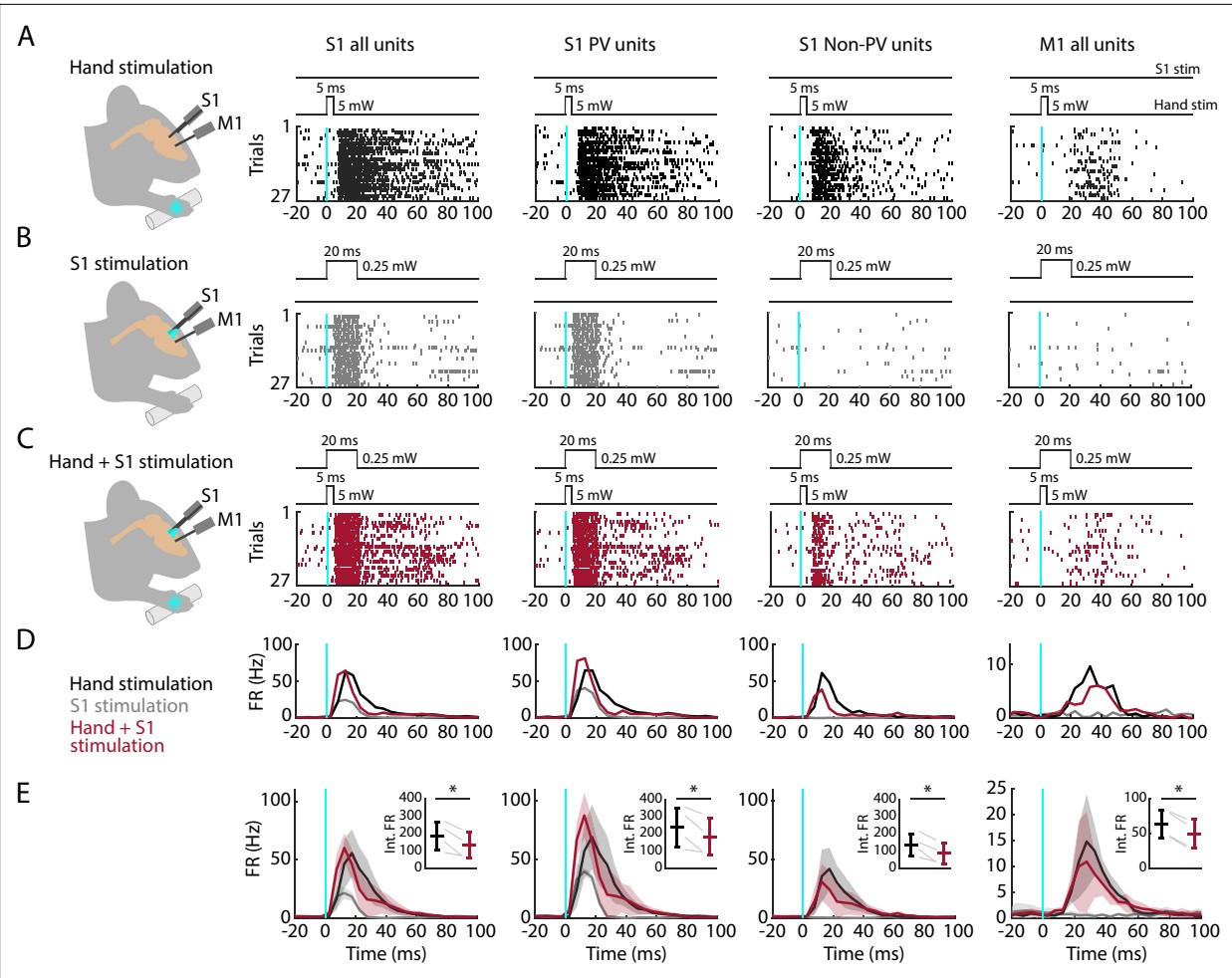

**Figure 8.** Partial silencing of S1 reduces M1 responses to hand stimulation. (**A**) First column: Experimental approach. Schematic depicts recording in S1 while simultaneous optogenetic stimulation of the hand of an awake head-fixed mouse (PV-Cre × Ai32). Example segment of a recording, showing raster plots of units responsive to hand stimulation (27 trials total). The schematic above the plot depicts the timing and parameters of stimulation (5 ms duration, 1 s inter-stimulus interval, 5 mW light intensity at the fiber tip, 910 μm core diameter). Second column: Same but for S1 opto-tagged stimulus-responsive parvalbumin (PV) units. Third column: Same but for S1 stimulus-responsive non-PV units. Fourth column: Same but for M1 units responsive to hand stimulation. (**B**) Same stimulus-responsive units as in A during focal optogenetic stimulation of PV neurons in S1 (27 trials total, 1 s inter-stimulus interval, 20 ms duration, 0.25 mW light intensity at the fiber tip, 105 μm core diameter). (**C**) Same stimulus-responsive units as in A during simultaneous hand and focal S1 photostimulation. (**D**) Average (mean) peristimulus time histogram (PSTH) across S1 units responsive to hand stimulation (first column), parsed into PV (second column), and non-PV units (third column) along with M1 activity (fourth column) aligned to the onset of the stimulation (black: hand, gray: cortex, maroon: hand + S1 stimulation) for the same example recording in (**A–C**). (**E**) Grand average (mean ± s.d.) PSTHs across recordings (4 recordings from 4 mice). Inset: Average activity (integral of FR) 15–50 ms after the stimulation onset (hand versus hand + S1 stimulation) along with the group comparison (paired $t$-Student test, S1 units: $t_3 = 4$, \*: p = 0.02; PV units: $t_3 = 3.2$, \*: p = 0.049; non-PV units: $t_3 = 4.6$, \*: p = 0.02; M1 units: $t_3 = 11.7$, \*: p = 0.001).

the activity in M1 was reduced (*Figure 8C, D*). A significant reduction of M1 activity was consistently observed within and across experiments (4 recordings from 4 mice; *Figure 8E*; *Table 2*). When S1 was partially silenced during hand stimulation, responses in S1 fell to 69.7% of control levels overall, with reduced activity of both PV (to 74.2%) and non-PV units (to 58.8%). At the same time, M1 responses also fell, to 76.2% overall. These results thus indicate that the responsiveness of M1 to peripheral stimulation of the hand depends at least in part on propagation of sensory-evoked activity via S1.

## Discussion

To characterize the propagation of somatosensory activity along the previously characterized fore-limb S1 and M1 cortical circuits in the mouse (*Yamawaki et al., 2021*), we developed an optogenetic

**Table 2.** Statistical results for the S1 silencing experiments.

| | Hand stim. (Int. FR) | Hand + S1 stim. (Int. FR) | (Hand + S1 stim.)/hand (%) | W | p |
|---|---|---|---|---|---|
| **S1** | | | | | |
| *All units* | | | | | |
| Mouse 1 | 271.1 | 225.7 | 83.3% | 5.2e2 | $2 \times 10^{-6}$ |
| Mouse 2 | 155.2 | 76 | 49.0% | 142.5 | $3 \times 10^{-8}$ |
| Mouse 3 | 94.1 | 75.1 | 79.8% | 793.5 | 0.002 |
| Mouse 4 | 228.4 | 165.2 | 72.3% | 428.5 | $7 \times 10^{-9}$ |
| Mean ± s.d. | 187.2 ± 78.4 | 135.5 ± 73.5 | 69.7 ± 1.2% | n/a | n/a |
| *PV units* | | | | | |
| Mouse 1 | 355.6 | 317.6 | 89.3% | 129.5 | 0.02 |
| Mouse 2 | 183.4 | 100.7 | 54.9% | 81.5 | $8 \times 10^{-5}$ |
| Mouse 3 | 111.4 | 96 | 86.2% | 377 | 0.047 |
| Mouse 4 | 308.9 | 222.2 | 71.9% | 92 | $2 \times 10^{-5}$ |
| Mean ± s.d. | 239.8 ± 112.3 | 184.1 ± 106.4 | 74.2 ± 1.2% | n/a | n/a |
| *Non-PV units* | | | | | |
| Mouse 1 | 216.4 | 166.3 | 76.8% | 125 | $2 \times 10^{-5}$ |
| Mouse 2 | 111.7 | 38 | 34.0% | 4 | $7 \times 10^{-5}$ |
| Mouse 3 | 66.2 | 41.5 | 62.7% | 79.5 | 0.01 |
| Mouse 4 | 151.6 | 110.8 | 73.2% | 125 | $7 \times 10^{-5}$ |
| Mean ± s.d. | 136.5 ± 63.7 | 89.15 ± 61.4 | 58.8 ± 1.4% | n/a | n/a |
| **M1** | | | | | |
| Mouse 1 | 81.9 | 71.4 | 87.2% | 148 | 0.01 |
| Mouse 2 | 46.1 | 31.2 | 67.7% | 47 | 0.01 |
| Mouse 3 | 79.8 | 63.9 | 80.1% | 365 | $8 \times 10^{-5}$ |
| Mouse 4 | 47.2 | 33.6 | 71.2% | 57.5 | 0.02 |
| Mean ± s.d. | 63.8 ± 19.8 | 50.0 ± 20.6 | 76.2 ± 1.1% | n/a | n/a |

Average (mean ± s.d.) activity (integral of firing rate, 'Int. FR') across stimulus-responsive units 15–50 ms after stimulation onset (of hand or hand + S1) along with the statistical intra-recording comparison (Wilcoxon's signed rank test, 4 recordings from 4 mice). All p-values remained significant after false discovery rate correction for multiple comparisons.

approach for delivering brief photostimuli to the hand while recording and in some cases manipulating population activity in the two cortical areas. The findings identify basic properties and mechanisms of population spiking dynamics in this model system for investigating neural mechanisms of sensorimotor integration in transcortical loops.

## Methodological considerations

The optogenetic method for peripheral stimulation used here, though artificial, closely mimics basic features of natural somatosensory stimulation, as shown in numerous prior studies involving related approaches (*O'Connor et al., 2013*; *Millard et al., 2015*; *Prsa et al., 2019*; *Emanuel et al., 2021*; *Lehnert et al., 2021*; *Schorscher-Petcu et al., 2021*). Like electrical stimulation, which is also used in somatosensory experiments, a key advantage is the ability to deliver temporally precise (i.e., brief) stimuli to the hand. However, electrical stimulation non-specifically activates somatosensory afferents, including pain fibers, and while tolerated by human subjects usually requires general anesthesia for studies in non-human mammalian model species (*Zaforas et al., 2024*).

Another advantage of the optogenetic approach is the ability to restrict labeling to particular somatosensory afferents of interest. A potential disadvantage of the particular approach used here, based on using a Cre line to drive expression, is that ChR2 may be expressed in the cortex as well, potentially constraining options for cortical optogenetic manipulations. However, in this case, we specifically chose the PV-Cre line because ChR2 is also expressed in cortical PV neurons, enabling cortical opto-tagging and silencing in the same experiments. A related point is that the particular Cre line used here (PV-Cre) is one that labels proprioceptor afferents (*Hippenmeyer et al., 2005*; *Wu et al., 2021*); other lines are available for labeling diverse somatosensory subtypes of DRG neurons (*Neubarth et al., 2020*; *Sharma et al., 2020*; *Wu et al., 2021*).

Our approach relied on repeated stimulation of the hand as it rested on the bar during each ~30 s block of trials, which was unproblematic in these experiments where the goal was to broadly stimulate a large but not necessarily identical population of mechanoreceptors distributed across the palm. However, for other experimental goals such as receptive field mapping, this configuration would need to be modified.

We primarily analyzed activity based on pooling of multi- and single units, as our main aim was to analyze the response modulation in S1 and M1 to the hand stimulation. This enabled more extensive exploration of the data, as the fraction of single units recorded was low. Additionally, basal firing rates were low, reflecting the quiescent behavioral state, as mice were required to remain still with their hand resting on the light-delivery bar during recordings. An acknowledged caveat, therefore, is that this approach is problematic for characterizing absolute firing rates.

## M1 responses reflect S1→M1 corticocortical activity

To what extent do the evoked responses reflect propagation of activity along S1→M1 corticocortical circuits (*Yamawaki et al., 2021*)? In principle, other pathways to M1 could be involved. For example, somatosensory thalamus projects not only to S1 but also S2, which projects to M1 (*Suter and Shepherd, 2015*). However, thalamocortical projections to S2 arise mainly from caudal whisker-related thalamic nuclei, while the forelimb-related pathways, including those for proprioception, are concentrated in rostral nuclei projecting to S1 (*Francis et al., 2008*; *Alonso et al., 2023*; *Rubio-Teves et al., 2024*). As another possibility, activity in S1 might drive M1 via thalamus (*Mo and Sherman, 2019*). However, in the mouse hand/forelimb system, cortico-thalamo-cortical (CTC) pathways are primarily organized as recurrent loops (*Guo et al., 2020*), which may recurrently drive corticocortical activity (*Shepherd and Yamawaki, 2021*). The reduction in M1 responses when S1 activity was partially silenced directly implicates S1 in driving M1 responses, and the non-PV silenced neurons in S1 likely included corticocortical neurons. Technical limitations, relating to the proximity of M1 and S1 and the spatial resolution of focal silencing, precluded even stronger silencing of S1 activity. Although residual M1 activity was most likely driven by remaining upstream activity within forelimb S1, other, albeit more circuitous, pathways may secondarily contribute. Overall, however, our findings indicate that M1 responses arise primarily from feedforward propagation of activity along S1→M1 circuits.

## Response latencies reflect fast subcortical and slow corticocortical circuits

In S1, the evoked activity developed quickly following hand stimulation, with onset latencies of 15 ms on average, consistent with those found previously using related approaches (*Emanuel et al., 2021*). For comparison, in the rat, whisker deflection generates cortical responses with an even shorter latency of ~10 ms (*Ahissar et al., 2000*), consistent with the shorter pathway distance for the whisker-barrel system. We estimated the effective propagation speed along the ascending pathway, from the periphery to S1, to be 3 m/s. Interestingly, in humans the hand-to-S1 latency is similarly short, ~20 ms, with correspondingly much higher conduction velocities for the ascending pathways (*Desmedt and Cheron, 1980*).

In M1, the evoked activity also developed quickly, lagging S1 by 10 ms. However, although the latencies were brief, the effective corticocortical propagation speed was slow, estimated at 0.14 m/s. This speed is comparable to the conduction velocity of thin unmyelinated cortical axons, estimated to be ~0.2 m/s (*Raastad and Shepherd, 2003*), and to interareal propagation speeds in other corticocortical circuits in mouse cortex, estimated to be approximately 0.3 m/s (*Li et al., 2018*). Both the hand-to-S1 and S1-to-M1 pathways are polysynaptic, and the estimated propagation speeds thus

reflect multiple biophysical properties (axonal conduction velocities, synaptic transmission delays, and more).

This 20-fold difference in subcortical versus corticocortical propagation speeds is consistent with previous characterizations of the synaptic circuit organization of the same pathways as 'streamlined' for the subcortical ascending pathways to S1 and 'densely polysynaptic' for the corticocortical pathway from S1 to M1 (*Yamawaki et al., 2021*). Thus, sensorimotor transformations in this system appear designed for signal propagation that is fast from periphery to the cortex, then slows drastically from S1 to M1 (and other areas). The slowness of corticocortical dynamics in this system accords with the idea that speed-limited sensorimotor transformations may be a general property of neural systems (*Zheng and Meister, 2025*).

## Response amplitudes and durations reflect intrinsic mechanisms of termination and attenuation

In M1, the initial peak response amplitudes were substantially attenuated relative to S1, where amplitudes were high, far above the low baseline firing rates. In both areas, responses were self-terminating, with brief durations. Response profiles resemble the 'packet'-like activity patterns previously described in diverse circuits, which have been proposed as fundamental units of information processing in neural systems (*Luczak et al., 2015*). They also closely resemble

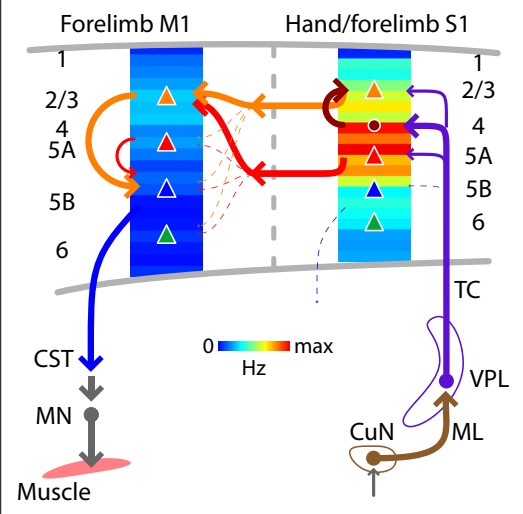

**Figure 9.** Schematic summary. Comparison of evoked cortical dynamics and synaptic circuit organization along the hand/forelimb-related transcortical loop through S1 and M1. The illustration combines the schematic summary from *Yamawaki et al., 2021*, which depicts the major local and corticocortical excitatory synaptic circuit connections along the loop, together with our current results, which show the laminar profiles of population spiking activity across the cortical depth in each area. The profiles show the same data as in *Figure 5A, B*, normalized to the maximum value in S1 plotted as heatmaps. CST: corticospinal tract, CuN: cuneate nucleus, ML: medial lemniscus, MN: motor neurons, TC: thalamocortical axons, VPL: ventral posterolateral nucleus of the thalamus. Adapted from Figure 7 of *Yamawaki et al., 2021*.

those of optogenetically evoked activity patterns observed in higher-order corticocortical pathways (*Li et al., 2018*).

Responses were not only self-terminating but also self-extending, in the sense that their durations (tens of milliseconds) were much longer than the peripheral stimulus that evoked them (5 ms). At the cortical level, this temporal dispersion of the responses did not accumulate going from S1 to M1, reflecting tight control and termination of excitation along the corticocortical pathway.

Response termination likely involves multiple mechanisms, given the importance of preventing runaway excitation. For example, cellular-level mechanisms such as spike frequency adaptation rapidly attenuate firing of mouse S1 and M1 neurons (*Hattox and Nelson, 2007*; *Suter et al., 2013*). Circuit-level mechanisms include the strong activation of cortical inhibitory interneurons by somatosensory input (considered later).

## Laminar activity reflects S1→M1 circuit organization

In S1, the laminar profile of evoked responses was broad, with a focus on middle laminar zones corresponding to layers 4 and 5A. This pattern is generally consistent with thalamocortical connectivity in forelimb S1 (*Yamawaki et al., 2021*), and with prior studies of homologous thalamocortical connections and dynamics in the whisker-barrel system (*de Kock et al., 2007*; *Constantinople and Bruno, 2013*; *Sermet et al., 2019*; *Yu et al., 2019*). In M1, previous results from slice-based analysis of S1→M1 connectivity show strongest excitatory synaptic input to layer 2/3 neurons, with little or no input to corticospinal and other deeper layer neurons (*Yamawaki et al., 2021*). Consistent with this, we found strongest activation in laminar zones corresponding to layers 2/3 and 5A. A small dip between these

zones, likely corresponding to layer 4, accords with other previous results showing a relatively paucity of corticocortical input to M1 layer 4 neurons (*Yamawaki et al., 2014*). Activity was low across deeper zones corresponding to layers 5B and 6. Thus the laminar profile of evoked activity largely matches that of the synaptic connectivity in the underlying corticocortical S1→M1 pathway (*Figure 9*).

## Triphasic S1 responses reflect mass activation of PV neurons

The expression of ChR2 in cortical PV neurons enabled identification of these neurons by opto-tagging during the hand-stimulation experiments, and thus quantification of their roles in the evoked responses in S1. Prior studies in S1 have shown that somatosensory stimulation potently activates cortical inhibitory interneurons, particularly PV-type neurons, with stronger and faster (shorter latency) recruitment compared to excitatory neurons (*Simons, 1978*; *Zhu and Connors, 1999*; *Bruno and Simons, 2002*; *Swadlow, 2003*; *Gabernet et al., 2005*; *Cruikshank et al., 2010*; *Yu et al., 2019*). Similarly, here we found that PV neurons in S1 are massively recruited by hand stimulation, both in terms of the number (approximately equal to that of non-PV units) and the firing rate (approximately double that of non-PV units). Indeed, most (~76%) of the evoked spikes in S1 cortex during the initial peak response were from PV neurons. PV units were moreover recruited slightly earlier (i.e., with shorter response latencies) than non-PV units. We did not study PV units in M1 in response to hand stimulation, but prior studies indicate similarly strong, short-latency recruitment of fast-spiking neurons in hand/forelimb M1 as well (*Murray and Keller, 2011*).

PV units remained involved in the ensuing suppression and rebound phases as well. During suppression, PV spiking fell, rather than increased; thus, the overall suppression could not be ascribed to ongoing firing of these inhibitory interneurons, but likely reflects other mechanisms, such as slow kinematics of GABAergic responses. During rebound, the contribution of PV spiking to the overall activity was again high compared to non-PV units, which in this case reflected their relatively greater recruitment rather than higher firing rates. These findings are consistent with prior characterizations of peak-suppression-rebound responses in S1 (*Chapin et al., 1981*; *Zhu and Connors, 1999*; *Sreenivasan, 2016*).

Selective activation of PV units alone in either S1 or M1 evoked a sequence of suppression and rebound activity similar to that evoked by hand stimulation alone, indicating a prominent role of PV activity in shaping the triphasic cortical responses. Focal activation of PV neurons, used here for the purpose of mechanistically probing cortical somatosensory responses, is also commonly used for the purpose of transient focal silencing (*Li et al., 2019*). Thus, the rebound activity studied here as part of the triphasic somatosensory response is closely related to the rebound excitation that can occur as an unwanted confounding side effect of focal activation of interneurons (*Wiegert et al., 2017*; *Li et al., 2019*). Our findings emphasize that this is an inherent property of the cortex, elicitable not only by peripheral sensory input but also intracortically by PV activation or electrical stimulation, and likely involving intrinsic electrophysiological mechanisms as well as local cortical and CTC circuits (*Grenier et al., 1998*; *Claar et al., 2023*; *Kumaravelu and Grill, 2024*; *Russo et al., 2024*). Forelimb-related S1 and M1 in the mouse both form strongly recurrent CTC circuits with thalamus (*Guo et al., 2018*; *Guo et al., 2020*).

Pairing selective activation of PV units in S1 with simultaneous hand stimulation caused the evoked somatosensory responses in both S1 and M1 to be attenuated. These findings suggest a role for PV neurons in S1 in regulating the extent to which S1 activity influences M1 activity. More generally, the findings obtained by opto-tagging PV neurons in S1 indicate a prominent role for these neurons in shaping both the local S1 and downstream M1 responses to somatosensory stimulation of the hand.

## Perspectives

The S1-to-M1 transcortical loop – the longest of several sensorimotor loops embedded along the neuraxis to enable somatosensory feedback control over a range of time scales – may be particularly important for mediating rapid adjustments to ongoing somatosensory information in the context of current and planned cortical motor commands (*Reschechtko and Pruszynski, 2020*). Our findings in the mouse provide a basis for comparisons to homologous systems, particularly the rodent vibrissal and primate hand/forelimb circuits; as alluded to above, these can exhibit notable similarities as well as differences in latencies and other dynamic properties. Our characterizations of basic response parameters in mouse hand/forelimb circuits also provide a framework for relating dynamic spiking

activity to the underlying cells and their synaptic connections in this system *Yamawaki et al., 2021*; as a first step, we explored the use of optogenetic manipulations of PV neurons. Additionally, building on the rapidly growing knowledge about cellular connectivity and behavior-related dynamics in 'top–down' corticocortical circuits (*Petreanu et al., 2009*; *Petreanu et al., 2012*; *Xu et al., 2012*; *Lee et al., 2013*; *Zagha et al., 2013*; *Kinnischtzke et al., 2014*; *Manita et al., 2015*), the stimulation and recording paradigms developed here could be adapted to investigate how inputs from higher-order motor areas such as secondary motor cortex (M2, or rostral forelimb area) interact with ascending somatosensory activity to modulate M1 output during goal-directed behaviors.

# Materials and methods

**Key resources table**

| Reagent type (species) or resource | Designation | Source or reference | Identifiers | Additional information |
|---|---|---|---|---|
| Strain, strain background (*M. musculus*) | PV-Cre or B6.129P2-*Pvalb*<sup>tm1(cre)Arbr</sup>/J | Jackson Laboratory (*Hippenmeyer et al., 2005*) | #017320 RRID:IMSR_JAX:017320 | |
| Strain, strain background (*M. musculus*) | Ai32 or B6.Cg-*Gt(ROSA)26Sor*<sup>tm32(CAG-COP4*H134R/EYFP)Hze</sup>/J | Jackson Laboratory (*Madisen et al., 2012*) | #024109 RRID:IMSR_JAX:024109 | |
| Strain, strain background (*M. musculus*) | Ai14 or B6.Cg-Gt(ROSA)26Sor<sup>tm14(CAG-tdTomato)Hze</sup>/J | Jackson Laboratory (*Madisen et al., 2010*) | #007914 RRID:IMSR_JAX:007914 | |
| Recombinant DNA reagent | pAAVretro-CAG-GFP | Addgene | #37825 RRID:Addgene_37825 | |
| Recombinant DNA reagent | AAVretro-CAG-tdTomato | Addgene | #59462 RRID:Addgene_59462 | |
| Chemical compound, drug | NBQX disodium salt | Tocris | #1044 PubChemCID: 3272523 | |
| Chemical compound, drug | Gabazine | Tocris | #1262 PubChemCID:107895 | |
| Software, algorithm | Open Ephys | Open Ephys | #021624 RRID:SCR_021624 | |
| Software, algorithm | MATLAB | http://www.mathworks.com/products/matlab/ | #001622 RRID:SCR_001622 | |
| Software, algorithm | KiloSort | https://github.com/cortex-lab/Kilosort (*Pachitariu et al., 2016*; *Steinmetz et al., 2021*; *Pachitariu, 2024*) | #016422 RRID:SCR_016422 | v3 |
| Software, algorithm | Wavesurfer | https://wavesurfer.janelia.org/ | | v0.945 |
| Software, algorithm | Phy | https://github.com/cortex-lab/phy | | |
| Other | Vybrant DiI Cell-Labeling Solution | Invitrogen | Catalog #: V22885 | See 'Probe track localization' |

## Animals

Experimental studies on mice followed protocols approved by the Institutional Animal Care and Use Committee of Northwestern University (#IS00019019) and fully complied with the animal welfare guidelines of the National Institutes of Health and the Society for Neuroscience. This study primarily used two mouse lines: the PV-Cre driver line of mice (B6.129P2-*Pvalb*<sup>tm1(cre)Arbr</sup>/J; RRID:IMSR_JAX:017320) (*Hippenmeyer et al., 2005*), which express Cre in peripheral mechanoreceptor afferents, particularly in proprioceptors, and also express Cre in cortical interneurons, particularly in PV-type cells; and, the Ai32 reporter line of mice (B6.Cg-*Gt(ROSA)26Sor*<sup>tm32(CAG-COP4*H134R/EYFP)Hze</sup>/J; RRID:IMSR_JAX:024109) (*Madisen et al., 2012*), which expresses a ChR2-EYFP fusion protein in Cre-expressing cells. Homozygous mice of each line were crossed to generate PV-Cre × Ai32 offspring. In a subset of experiments, to image PV neurons in brain sections, the PV-Cre mice were crossed with Ai14 (tdTomato) reporter

mice (B6.Cg-Gt(ROSA)26Sor$^{tm14(CAG-tdTomato)Hze}$/J; RRID:IMSR_JAX:007914) (*Madisen et al., 2010*). Mice were bred in-house and housed in groups on a 12-hr reverse light/dark cycle, with free access to food and water. Experiments were conducted during the dark phase of the light cycle. Adult mice of both sexes were used (12 male, 14 female). Mice were not selected based on sex and were used as they became available.

## Surgical procedures

Head-bar implantation was performed as previously described (*Barrett et al., 2022*). Briefly, mice under deep isoflurane anesthesia were placed in a stereotaxic frame. The scalp and periosteum were removed to expose the cranium, a titanium bar was cemented over lambda, and the exposed cranium was covered with dental cement. For analgesia, mice were given 0.3 mg/kg buprenorphine preoperatively, and 1.5 mg/kg meloxicam both immediately and one day postoperatively.

In a subset of experiments, corticospinal neurons were labeled by injecting a retrograde AAV (pAAVretro-CAG-GFP, 37825-AAVrg; Addgene, Watertown, MA) (*Tervo et al., 2016*) into the spinal cord at the same time as head-bar mounting, following previously described methods (*Barrett et al., 2022*).

## Electrophysiological recordings

In vivo electrophysiological recordings from awake mice were performed similarly to previous studies (*Barrett et al., 2022*). Mice were habituated first to handling by the experimenter and then to head fixation for 3 days prior to recording. Then, under deep isoflurane anesthesia, a dental drill was used to open one or more craniotomies over the cortical areas to be recorded. Recordings were made using 64-channel silicon probes (model A1x64-Poly2-6mm-23s-160-A64, Z-coated; NeuroNexus, Ann Arbor, MI) and opto-probes (model A1x64-Poly2-6mm-23s-160-OA64LP, Z-coated, core diameter of the coupled fiber 105 μm, NA 0.22, termination 200 μm above top channel; NeuroNexus), with ~1 MΩ impedance, and 23 μm vertical spacing in a horizontally staggered configuration (30 μm horizontal spacing, total recording length 1449 μm) (*Figure 1—figure supplement 2*).

Probes were mounted on a linear translator (MTS25-Z8, ThorLabs, Inc) on a 3-axis manipulator (1U RACK, Scientifica, Uckfield, UK). Probes were slowly (5 μm/s) inserted under software control (Kinesis, Thorlabs, Inc) into the cortical target areas to a nominal depth of 1500 μm. Target coordinates for forelimb M1 were 0.25 mm anterior–posterior (AP), 1.35 mm medial–lateral (ML); those for forelimb S1 were 0.0 mm AP, 2.4 mm ML (*Yamawaki et al., 2021*). A subgroup of mice with both corticospinal and probe track labeling was used to further confirm accurate targeting of these coordinates, based on prior characterizations of corticospinal neuron distributions extending from the hand/forelimb-related subregions of M1 medially to S1 laterally (*Ueno et al., 2018*; *Yamawaki et al., 2021*). For each mouse, recordings were made over 1 or 2 days from S1 and M1 of one or both hemispheres. After each recording session the probes were removed and the craniotomy was re-sealed with Kwik-Sil (World Precision Instruments, Sarasota, FL). Mice were transcardially perfused with 4% paraformaldehyde in phosphate buffered saline after finishing all experiments and the brains were processed to localize the probe tracks as previously described (*Barrett et al., 2022*).

Data acquisition hardware included RHD2132 headstages (Intan Technologies, Los Angeles, CA) and an Open Ephys data acquisition board (OEPS-6501, Open Ephys, Lisbon, Portugal). Signals were hardware bandpass-filtered (2.5 Hz to 7.6 KHz) at acquisition (with no additional software filtering) and sampled at 30 KHz. Software control was with Open Ephys.

## Spike sorting

Open Ephys binary files were converted to raw format using Matlab (The MathWorks, Natick, MA). As previously described (*Barrett et al., 2022*), spikes were detected and sorted using Kilosort (*Pachitariu et al., 2016*; *Steinmetz et al., 2021*) and verified using phy (https://github.com/cortex-lab/phy, copy archived at *cortex-lab, 2024*) using standard methods to reject artifactual units (waveforms spanning more than three adjacent channels, or atypical waveform shapes) and identify single units (refractory period; <1% of spikes within 1 ms) and multi-units (all others). Multi-units on the same channel with similar waveform shapes were merged. If two or more sorted single units with similar waveform shapes were detected on the same channel, and their cross-correlogram had no spikes within in ±1 ms, these were assumed to be a single unit erroneously split by Kilosort, and hence

manually merged (*Hall et al., 2021*). Single units and multi-units were pooled as 'active units'. Overall, approximately one-third of all sorted units were single units (30% in S1, 33% in M1); the proportion of all stimulus-responsive units that were single units was somewhat lower (14% in S1, 19% in M1). Only probe recordings with at least 15 active units were analyzed.

## Optogenetic photostimulation of the hand

The experimental apparatus was assembled on a vibration isolation table, and included a raised platform on which a 3D-printed hut was placed for the mouse to sit in, head-bar holders with screw clamps, and hand rest. For photostimulation of hand mechanoreceptors, the bar for the hand rest was fashioned out of a thick-walled plastic tube (diameter: 3.2 mm outside, 1.5 mm inside) by drilling a through-hole and inserting the exposed end of an optical fiber (910 μm core diameter, 0.22 NA, 1.5 m length, FG910UEC Patch cable, Thorlabs, Inc). The fiber was coupled to a blue LED (455 nm wavelength, M455F3 source, LEDD1B driver; Thorlabs, Inc). The bar was positioned so that the fiber tip faced the palm when mice placed their hand on the resting bar. With this configuration, light stimuli broadly illuminated the volar aspect of the hand. We visually monitored (and video recorded in a subset of experiments) the hand placement during each block of trials, and blocks in which the mouse repositioned the hand were discarded.

Photostimulation parameters were controlled by Wavesurfer software (v0.945, https://wavesurfer.janelia.org/). Light intensity at the fiber tip was calibrated with a power meter (PM100D, Thorlabs, Inc). Stimulus duration was 5 ms and intensity was 5 mW at the fiber tip. This intensity was chosen based on pilot experiments in which we varied the LED power, which showed that this intensity was reliably above the threshold for evoking robust responses in both S1 and M1 without evoking any visually detectable movements (as subsequently confirmed by videography). Photostimuli were applied to the hand contralateral to the cortical recordings. For each experiment, light pulses were delivered in blocks of 25 trials (except where noted otherwise) with an inter-trial period of 1 s. In addition, we also recorded responses when the mouse's hand was resting off the bar, either as a control or in combination with cortical photostimulation (described next).

## Optogenetic photostimulation of cortical PV neurons

In most cases, optogenetic activation of PV neurons in S1 or M1 was performed while mice had their hands off the light-delivery bar, using a sequence of 25 pulses of blue light delivered at the tip of the opto-probe. The probe was inserted as described above and coupled via patch cord (M63L01, Thorlabs, Inc) to the blue laser source (473 nm wavelength, MBL-III-473-100 mW, Opto Engine LLC), with an illumination intensity of 1 mW at the fiber tip, core diameter 105 μm. For a subset of silencing experiments, the optogenetic activation of PV neurons was interleaved with trials of hand stimulation or both hand and focal S1 stimulation while the mouse kept their hand resting on the light-delivery bar. The illumination intensity was reduced to 0.25 mW at the fiber tip. There was a total of 27 trials per condition in this case. In all cases, the stimulus duration for PV activation was 20 ms.

We checked the specificity of cortical labeling in the PV-Cre mice (*Figure 6—figure supplement 1*), as prior results have shown PV expression in a subset of excitatory neurons in layer 5 (*Palicz et al., 2024*). Anatomically, in cortical slices of PV-Cre x Ai14 (tdTomato) mice in which layer 5B corticospinal neurons were also labeled, we observed no double-labeled cells. Electrophysiologically, in cortical slices of PV-Cre × Ai32 mice in which corticospinal and corticocortical excitatory neurons were also labeled, recordings from these neurons showed no detectable photocurrents. In some neurons, we did detect evoked excitatory synaptic currents and/or potentials. However, these were weak, and tiny compared to the GABAergic inputs. Thus, in the experiments reported here, the dominant effect of cortical photostimulation in PV-Cre × Ai32 mice appears consistent with massive GABA-mediated inhibition.

## Analysis of stimulus-aligned data

For each active unit, spiking activity was binned in 5 ms bins, aligned to stimulus onset. PSTHs spanning 0.5 s before to 0.5 s after stimulation were constructed by averaging the stimulus-aligned binned firing rates over trials (generally 25). To detect stimulus-responsive units, the PSTHs were $z$-scored, and stimulus-responsive units were identified as those for which the maximum $z$-scored firing rate exceeded 2.5 s.d. above the baseline mean value (250 ms preceding the stimulus onset).

Analyses focused on stimulus-responsive units, except where otherwise indicated. Dynamic properties of each responsive unit's PSTH were extracted as follows. To analyze properties associated with the early main response, for each trial, we first baseline-subtracted the raw PSTHs using their baseline mean value. Peak amplitude was calculated as the maximum firing rate in the average baseline-subtracted PSTH, over a time window of 100 ms post-stimulus. The time bin of the maximum post-stimulus firing rate was taken as the peak latency. The time bin when the firing rate first exceeded 2.5 s.d. was considered as the onset latency. Response duration was computed as the time window from when the firing rate first exceeded 2.5 s.d. above baseline to when it first fell below 1 s.d above baseline.

To analyze properties associated with the post-peak suppression phase of the responses, we identified units with significant suppression: for each unit, we calculated the mean firing rate during this interval, between 110 and 170 ms (based on visual inspection of average PSTHs), and statistically compared this to the mean of the pre-stimulus baseline (Wilcoxon's signed rank test). As the response tended to rebound after this period, we identified units with significant rebound: for each unit, we calculated the rebound amplitude by finding the maximum of the PSTH during the rebound window, between 190 and 400 ms, and averaging from 20 ms before to 20 ms after this value, and statistically compared this with the mean of the pre-stimulus baseline.

For laminar analysis, we computed the PSTH for all units as above but sorted the units into bins of 5% of the normalized cortical depth, 20 bins total from 0 (pia) to 1 (boundary of cortex with the underlying white matter). The bin-averaged PSTHs were then calculated using the same approach as described above for the probe-averaged PSTHs.

For each experiment, once the parameters and PSTHs were computed for all responsive units, values were pooled and averaged. These averages were then pooled to obtain grand averages across experiments.

## Video motion analysis

To analyze potential stimulus-evoked movements, video recordings were taken from a cohort of three mice during photostimulation of the hand. Videos were captured under red light illumination with a high-speed CMOS-based monochrome video camera at 1000 frames per second (fps), 999.6 μs exposure time, and 1024 × 512 pixel field of view, as previously described (*Barrett et al., 2024*). Motion analysis was performed by taking the difference in pixel values for each frame from the average values of the same pixels in a 100-ms baseline preceding each stimulus, for at least 25 trials. The average difference in pixel values in a rectangular ROI surrounding the hand was then calculated over a post-stimulus period from 6 to 100 ms after each stimulus (the duration the LED was on was excluded from analysis due to the artifact caused by the illumination). If a mouse moved in any consistent manner during the post-stimulus interval, this would be reflected in a narrow range of average change in pixel values on each trial, but for all three mice the average change in pixel values was not significantly different from zero (Wilcoxon signed rank test, all p > 0.05, *Figure 1—figure supplement 1*; *Figure 1—video 1*).

## Probe track localization

As previously described (*Barrett et al., 2022*), recording sites were confirmed by coating the probes with a fluorescent dye (DiI, Vybrant Multicolor Cell Labeling Kit, Invitrogen, Carlsbad, CA) prior to recordings, harvesting and histologically processing the brains after experiments, and localizing the probe tracks in 100 μm brain sections under epifluorescence microscopy. In two cases, because probe tracks could not be recovered histologically, areal assignment was based on the targeted coordinates.

## Pathway morphometry

S1-to-M1 distance was calculated as the Euclidean distance between the S1 and M1 recording sites, localized based on probe tracks as described above. To estimate the hand-to-S1 distance we separately estimated the hand-to-cord and cord-to-cortex distances and summed these. The distance from the palm of the hand to the sixth cervical (C-6) spinal cord segment was estimated from morphometric measurements of cadaveric specimens (*n* = 5 mice). The distance from C-6 cord to S1 cortex was estimated from published MRI images (*Harrison et al., 2013*).

## Slice-based optogenetics and electrophysiology

In a subset of experiments we used standard slice-based methods to assess responses of electrophysiologically recorded pyramidal neurons to photostimulation of ChR2-expressing presynaptic axons in brain slices of forelimb S1 and M1, as previously described in detail (*Yamawaki et al., 2016*; *Yamawaki et al., 2021*). Briefly, the technical approach was essentially identical to that used previously to characterize excitatory connectivity in the same circuits (*Yamawaki et al., 2021*), adapted to examine excitatory and inhibitory synaptic inputs from ChR2-expressing PV neurons (in PV-Cre × Ai32 mice, as described above) to layer 2/3 and 5B pyramidal neurons. To facilitate positive identification of pyramidal neurons, retrograde AAV-tdTomato (AAVretro-CAG-tdTomato, 59462-AAVrg; Addgene, Watertown, MA) was injected into the spinal cord (to label corticospinal neurons, as described above) and in the contralateral cortex (to label intratelencephalic-type projection neurons). For additional details, see *Figure 6—figure supplement 1*.

## Statistical analysis

Group data are presented as mean ± s.d. unless otherwise indicated. For group comparisons, we primarily used nonparametric tests. Except where indicated otherwise, for paired samples Wilcoxon's signed rank was calculated with a minimum sample size of $n = 6$. Where the number of experiments was smaller than this, paired $t$-Student tests were calculated instead (*Figure 8E*, *Figure 7—figure supplement 1*), or within-animal Wilcoxon's signed rank tests were performed (*Table 2*), using the Benjamini–Hochberg procedure applied to the resulting p-values to control for the false discovery rate (*Benjamini and Hochberg, 1995*). The Friedman test was used for one-way repeated measures analysis of variance, with a minimum sample size of $n = 4$, and Dunn–Sidak as a post hoc multiple comparison test. For correlations, Spearman's coefficients were calculated and $t$-tested afterwards. In all cases significance was set at $p < 0.05$.

## Acknowledgements

We thank Megan Martin and Miraya Baid for technical assistance, and the Northwestern University Center for Advanced Microscopy (RRID:SCR_020996), supported by NCI CCSG P30 CA060553 awarded to the Robert H Lurie Comprehensive Cancer Center. National Institute of Neurological Disorders and Stroke (NS125594 and NS061963) – Gordon Shepherd.

## Additional information

### Funding

| Funder | Grant reference number | Author |
| --- | --- | --- |
| National Institute of Neurological Disorders and Stroke | NS125594 | Gordon MG Shepherd |
| National Institute of Neurological Disorders and Stroke | NS061963 | Gordon MG Shepherd |

The funders had no role in study design, data collection, and interpretation, or the decision to submit the work for publication.

### Author contributions

Daniela Piña Novo, Conceptualization, Resources, Data curation, Software, Formal analysis, Investigation, Visualization, Methodology, Writing – original draft, Writing – review and editing; Mang Gao, Conceptualization, Resources, Writing – review and editing; Rita Fischer, Investigation, Visualization; Louis Richevaux, Investigation, Visualization, Writing – review and editing; Jianing Yu, Conceptualization, Writing – review and editing; John M Barrett, Conceptualization, Resources, Software, Formal analysis, Supervision, Investigation, Methodology, Project administration, Writing – review and

editing; Gordon MG Shepherd, Conceptualization, Resources, Funding acquisition, Writing – original draft, Project administration, Writing – review and editing

## Author ORCIDs

Daniela Piña Novo (ID) https://orcid.org/0009-0003-1000-0805
Mang Gao (ID) http://orcid.org/0009-0002-0379-8655
Rita Fischer (ID) http://orcid.org/0009-0008-4709-8822
Louis Richevaux (ID) http://orcid.org/0000-0001-7837-3489
John M Barrett (ID) https://orcid.org/0000-0002-7524-6035
Gordon MG Shepherd (ID) https://orcid.org/0000-0002-1455-8262

## Ethics

Experimental studies on mice followed protocols approved by the Institutional Animal Care and Use Committee of Northwestern University (#IS00019019) and fully complied with the animal welfare guidelines of the National Institutes of Health and the Society for Neuroscience.

Reviewer #1 (Public review): https://doi.org/10.7554/eLife.105112.3.sa1
Reviewer #2 (Public review): https://doi.org/10.7554/eLife.105112.3.sa2
Reviewer #3 (Public review): https://doi.org/10.7554/eLife.105112.3.sa3
Author response https://doi.org/10.7554/eLife.105112.3.sa4

---

# Additional files

## Supplementary files

MDAR checklist

## Data availability

Data for this study is available at Zenodo at https://doi.org/10.5281/zenodo.14983570.

The following dataset was generated:

| Author(s) | Year | Dataset title | Dataset URL | Database and Identifier |
|---|---|---|---|---|
| Piña Novo D, Shepherd G, Barrett JM | 2025 | Dataset for "Cortical dynamics in hand/forelimb S1 and M1 evoked by brief photostimulation of the mouse's hand | https://doi.org/10.5281/zenodo.14983570 | Zenodo, 10.5281/zenodo.14983570 |

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
