## [Editor Report · eLife Assessment]

This work defines the response dynamics in forepaw-related cortical circuits of S1 and M1 following stimulation of peripheral mechanoreceptors in the mouse. In this revised version, the authors have addressed the reviewers' prior concerns. The results are **convincing** and present a **valuable** comparison to previously published work. This study has implications for understanding the interactions between primary somatosensory and motor cortex, required for active sensing, and will be of interest to scientists seeking to better understand the functions of somatosensory and motor circuits.

---

## [Referee Report · Reviewer #1 (Public review)]

Summary:

Building on previous in vitro synaptic circuit work (Yamawaki et al., eLife 10, 2021), Piña Novo et al. utilize an in vivo optogenetic-electrophysiological approach to characterize sensory-evoked spiking activity in the mouse's forelimb primary somatosensory (S1) and motor (M1) areas. Using a combination of a novel "phototactile" somatosensory stimuli to the mouse's hand and simultaneous high-density linear array recordings in both S1 and M1, the authors report evoked activity in S1 was biased to middle layers, whereas it was biased to upper layers in M1. They report that M1 responses are delayed and attenuated relative to S1. Further analysis revealed a 20-fold difference in subcortical versus corticocortical propagation speeds. They also find that PV interneurons in S1 are strongly recruited by hand stimulation, and their selective activation can produce a suppression and rebound response similar to "phototactile" stimuli. Silencing S1 through local PV cells was sufficient to reduce M1 response to hand stimulation, suggesting S1 may directly drive M1 responses.

Strengths:

The study was technically well done, with convincing results. The data presented are appropriately analyzed. The author's findings build on a growing body of both in vitro and in vivo work examining the synaptic circuits underlying the interactions between S1 and M1. The paper is well-written and illustrated. Overall, the study will be valuable to those interested in forelimb S1-M1 interactions.

Weaknesses:

The authors have addressed my concerns

---

## [Referee Report · Reviewer #2 (Public review)]

Summary:

Communication between sensory and motor corticies is likely to be important for many aspects of behavior, and in this study the authors carefully analyse neuronal spiking activity in S1 and M1 evoked by peripheral paw stimulation finding clear evidence for sensory responses in both cortical regions

Strengths:

The experiments and data analyses appear to have been carefully carried out and clearly represented.

Weaknesses:

The revised manuscript addressed the minor weaknesses I noted relating to the first submission.

---

## [Referee Report · Reviewer #3 (Public review)]

Summary:

This is a solid study of stimulus-evoked neural activity dynamics in the feedforward pathway from mouse hand/forelimb mechanoreceptor afferents to S1 and M1 cortex. The conclusions are generally well supported and match expectations from previous studies of hand/forelimb circuits by this same group (Yamawaki et al., 2021), from the well-studied whisker tactile pathway to whisker S1 and M1, and from the corresponding pathway in primates. The study uses the novel approach of optogenetic stimulation of PV afferents in the periphery, which provides an impulse-like volley of peripheral spikes, which is useful for studying feedforward circuit dynamics. These are primarily proprioceptors, so results could differ for specific mechanoreceptor populations, but this is a reasonable tool to probe basic circuit activation. Mice are awake but not engaged in a somatosensory task, which is sufficient for the study goals.

The main results are: (1) brief peripheral activation drives brief sensory-evoked responses at ~ 15 ms latency in S1 and ~25 ms latency in M1, which is consistent with classical fast propagation on the subcortical pathway to S1, followed by slow propagation on the polysynaptic, non-myelinated pathway from S1 to M1; (2) each peripheral impulse evokes a triphasic activation-suppression-rebound response in both S1 and M1; (3) PV interneurons carry the major component of spike modulation for each of these phases; (4) activation of PV neurons in each area (M1 or S1) drives suppression and rebound both in the local area and in the other downstream area; (5) peripheral-evoked neural activity in M1 is at least partially dependent on transmission through S1.

All conclusions are well-supported and reasonably interpreted. There are no major new findings that were not expected from standard models of somatosensory pathways or from prior work in the whisker system.

Strengths:

This is a well-conducted and analyzed study in which the findings are clearly presented. The optogenetic sensory afferent stimulation method is novel and is well-suited for studying feedforward circuit dynamics. This study provides important baseline knowledge from which studies of more complex sensorimotor processing can build.

There are no further recommendations for the authors.

---

## [Author Response]

The following is the authors’ response to the original reviews

**Reviewer #1 (Public review):**
Summary:Building on previous in vitro synaptic circuit work (Yamawaki et al., eLife 10, 2021), Piña Novo et al. utilize an in vivo optogenetic-electrophysiological approach to characterize sensory-evoked spiking activity in the mouse's forelimb primary somatosensory (S1) and motor (M1) areas. Using a combination of a novel "phototactile" somatosensory stimuli to the mouse's hand and simultaneous high-density linear array recordings in both S1 and M1, the authors report in awake mice that evoked cortical responses follow a triphasic peak-suppression-rebound pattern response. They also find that M1 responses are delayed and attenuated relative to S1. Further analysis revealed a 20-fold difference in subcortical versus corticocortical propagation speeds.They also report that PV interneurons in S1 are strongly recruited by hand stimulation. Furthermore, they report that selective activation of PV cells can produce a suppression and rebound response similar to "phototactile" stimuli. Lastly, the authors demonstrate that silencing S1 through local PV cell activation reduces M1 response to hand stimulation, suggesting S1 may directly drive M1 responses.Strengths:The study was technically well done, with convincing results. The data presented are appropriately analyzed. The author's findings build on a growing body of both in vitro and in vivo work examining the synaptic circuits underlying the interactions between S1 and M1. The paper is well-written and illustrated. Overall, the study will be useful to those interested in forelimb S1-M1 interactions.Weaknesses:Although the results are clear and convincing, one weakness is that many results are consistent with previous studies in other sensorimotor systems, and thus not all that surprising. For example, the findings that sensory stimulation results in delayed and attenuated responses in M1 relative to S1 and that PV inhibitory cells in S1 are strongly recruited by sensory stimulation are not novel (e.g., Bruno et al., J Neurosci 22, 10966-10975, 2002; Swadlow, Philos Trans R Soc Lond B Biol Sci 357, 1717-1727, 2002; Gabernet et al., Neuron 48, 315-327, 2005; Cruikshank et al., Nat Neurosci 10, 462-468, 2007; Ferezou et al., Neuron 56, 907-923, 2007; Sreenivasan et al., Neuron 92, 1368-1382, 2016; Yu et al., Neuron 104, 412-427 e414, 2019). Furthermore, the observation that sensory processing in M1 depends upon activity in S1 is also not novel (e.g., Ferezou et al., Neuron 56, 907-923, 2007; Sreenivasan et al., Neuron 92, 1368-1382, 2016). The authors do a good job highlighting how their results are consistent with these previous studies.

We thank the reviewer for the close reading of the manuscript and the many constructive comments and critiques. As the reviewer notes, there have been many prior studies of related circuits in other sensorimotor systems forming an important context for our study and findings, as we have tried to highlight. We appreciate the suggestions for additional relevant articles to cite.

Perhaps a more significant weakness, in my opinion, was the missing analyses given the rich dataset collected. For example, why lump all responsive units and not break them down based on their depth? Given superficial and deep layers respond at different latencies and have different response magnitudes and durations to sensory stimuli (e.g., L2/3 is much more sparse) (e.g., Constantinople et al., Science 340, 1591-1594, 2013; Manita et al., Neuron 86, 1304-1316, 2015; Petersen, Nat Rev Neurosci 20, 533-546, 2019; Yu et al., Neuron 104, 412-427 e414, 2019), their conclusions could be biased toward more active layers (e.g., L4 and L5). These additional analyses could reveal interesting similarities or important differences, increasing the manuscript's impact. Given the authors use high-density linear arrays, they should have this data.

We have analyzed the activity patterns as a function of cortical depth, and now include these results in the manuscript as suggested. The key new finding is that the M1 responses are strongest in upper layers, consistent with expectations based on the excitatory corticocortical synaptic connectivity characterized previously. Changes to the manuscript include new figures (Figure 5; Figure 5 - figure supplement 1), which we explain (Methods: page 14, lines 618-621), describe (new Results section: pages 4-5, lines 183-189), comment on (Discussion: page 9, lines 378-391), and summarize the significance of (Abstract: page 1, lines 22-24). In addition, we incorporated the new laminar analysis into a summary schematic (Figure 9). We thank the reviewer for suggesting this analysis.

Similarly, why not isolate and compare PV versus non-PV units in M1? They did the photostimulation experiments and presumably have the data. Recent in vitro work suggests PV neurons in the upper layers (L2/3) of M1 are strongly recruited by S1 (e.g., Okoro et al., J Neurosci 42, 8095-8112, 2022; Martinetti et al., Cerebral cortex 32, 1932-1949, 2022). Does the author's data support these in vitro observations?

These experiments were relatively complex and M1 optotagging was not routinely included in the stimulus and acquisition protocol. Therefore, we don’t have sufficient data for this analysis. We plan to address this in future studies.

It would have also been interesting to suppress M1 while stimulating the hand to determine if any part of the S1 triphasic response depends on M1 feedback.

We agree that this is of interest but consider this to be outside the scope of the current study.

I appreciate the control experiment showing that optical hand stimulation did not evoke forelimb movement. However, this appears to be an N=1. How consistent was this result across animals, and how was this monitored in those animals? Can the authors say anything about digit movement?

We have performed additional experiments to address this point. A constraint with EMG is that it is limited to the muscle(s) one chooses to record from, and it is difficult to implant tiny muscles of the hand. Therefore, for this analysis, we used kilohertz videography as a high-sensitivity method for movement surveillance across the hand. Hand stimulation did not evoke any detectable movements. Changes in the manuscript include: revised Figure 1 - figure supplement 1; supplementary Figure 1 - video 1; and associated text edits in the Methods (page 13, line 557; page 14, lines 626-639) and Results sections (page 2, lines 84-85).

A light intensity of 5 mW was used to stimulate the hand, but it is unclear how or why the authors chose this intensity. Did S1 and M1 responses (e.g., amplitude and latency) change with lower or higher intensities? Was the triphasic response dependent on the intensity of the "phototactile" stimuli?

As we now say in the Methods > Optogenetic photostimulation of the hand section (page 13, lines 562-565), “This intensity was chosen based on pilot experiments in which we varied the LED power, which showed that this intensity was reliably above the threshold for evoking robust responses in both S1 and M1 without evoking any visually detectable movements (as subsequently confirmed by videography)”.

**Reviewer #2 (Public review):**
Summary:Communication between sensory and motor cortices is likely to be important for many aspects of behavior, and in this study, the authors carefully analyse neuronal spiking activity in S1 and M1 evoked by peripheral paw stimulation finding clear evidence for sensory responses in both cortical regionsStrengths:The experiments and data analyses appear to have been carefully carried out and clearly represented.Weaknesses:(1) Some studies have found evidence for excitatory projection neurons expressing PV and in particular some excitatory pyramidal cells can be labelled in PV-Cre mice. The authors might want to check if this is the case in their study, and if so, whether that might impact any conclusions.

Thank you for pointing this out. The prior studies suggest it is mainly a subset of layer 5B excitatory neurons that may express PV. We checked this in two ways. Anatomically, we did not find double-labeling. An electrophysiology assay showed that, although some evoked excitatory synaptic input could be detected in some neurons, these inputs were very weak. Results from these assays are shown in new Figure 6 - figure supplement 1, with associated text edits in the Methods (page 11, lines 469-471; page 15, lines 657-668) and Results (page 5, lines 198-199) sections.

(2) I think the analysis shown in Figure S1 apparently reporting the absence of movements evoked by the forepaw stimulation could be strengthened. It is unclear what is shown in the various panels. I would imagine that an average of many stimulus repetitions would be needed to indicate whether there is an evoked movement or not. This could also be state-dependent and perhaps more likely to happen early in a recording session. Videography could also be helpful.

As noted above, we have performed additional experiments to address this.

(3) Some similar aspects of the evoked responses, including triphasic dynamics, have been reported in whisker S1 and M1, and the authors might want to cite Sreenivasan et al., 2016.

Thank you for pointing this out; we now cite this article (page 1, line 46; page 10, line 415).

**Reviewer #3 (Public review):**
Summary:This is a solid study of stimulus-evoked neural activity dynamics in the feedforward pathway from mouse hand/forelimb mechanoreceptor afferents to S1 and M1 cortex. The conclusions are generally well supported, and match expectations from previous studies of hand/forelimb circuits by this same group (Yamawaki et al., 2021), from the well-studied whisker tactile pathway to whisker S1 and M1, and from the corresponding pathway in primates. The study uses the novel approach of optogenetic stimulation of PV afferents in the periphery, which provides an impulselike volley of peripheral spikes, which is useful for studying feedforward circuit dynamics. These are primarily proprioceptors, so results could differ for specific mechanoreceptor populations, but this is a reasonable tool to probe basic circuit activation. Mice are awake but not engaged in a somatosensory task, which is sufficient for the study goals.The main results are:(1) brief peripheral activation drives brief sensory-evoked responses at ~ 15 ms latency in S1 and ~25 ms latency in M1, which is consistent with classical fast propagation on the subcortical pathway to S1, followed by slow propagation on the polysynaptic, non-myelinated pathway from S1 to M1;(2) each peripheral impulse evokes a triphasic activation-suppression-rebound response in both S1 and M1;(3) PV interneurons carry the major component of spike modulation for each of these phases; (4) activation of PV neurons in each area (M1 or S1) drives suppression and rebound both in the local area and in the other downstream area;(5) peripheral-evoked neural activity in M1 is at least partially dependent on transmission through S1.All conclusions are well-supported and reasonably interpreted. There are no major new findings that were not expected from standard models of somatosensory pathways or from prior work in the whisker system.Strengths:This is a well-conducted and analyzed study in which the findings are clearly presented. This will provide important baseline knowledge from which studies of more complex sensorimotor processing can build.Weaknesses:A few minor issues should be addressed to improve clarity of presentation and interpretation:(1) It is critical for interpretation that the stimulus does not evoke a motor response, which could induce reafference-based activity that could drive, or mask, some of the triphasic response. Figure S1 shows that no motor response is evoked for one example session, but this would be stronger if results were analyzed over several mice.

As noted above, we have performed additional experiments to address this point.

(2) The recordings combine single and multi-units, which is fine for measures of response modulation, but not for absolute evoked firing rate, which is only interpretable for single units. For example, evoked firing rate in S1 could be higher than M1, if spike sorting were more difficult in S1, resulting in a higher fraction of multi-units relative to M1. Because of this, if reporting of absolute firing rates is an essential component of the paper, Figs 3D and 4E should be recalculated just for single units.

Thank you for noting this. Although the absolute firing rates are not essential for the main findings or conclusions (which as noted focus on response modulations and relative differences) we agree that analyzing the single-unit response amplitudes is useful. Therefore, changes in the manuscript now include: revised Figure 3, and associated text edits in the Methods (page 12, lines 543-545), Results (page 3, lines 115-119), and Discussion (page 7, lines 305-311) sections.

(3) In Figure 5B, the average light-evoked firing rate of PV neurons seems to come up before time 0, unlike the single-trial rasters above it. Presumably, this reflects binning for firing rate calculation. This should be corrected to avoid confusion.

Yes, this reflects the binning. We agree that this is potentially confusing and have removed these average plots below the raster plots, as the rasters alone suffice to demonstrate the result (i.e., that PV units are strongly activated and thus tagged by optogenetic stimulation). Changes are now reflected in revised Figure 6.

(4) In Figure 6A bottom, please clarify what legends "W. suppression" and "W. rebound" mean.

In the figure plot legends, the “W.” has been removed. Changes are now reflected in revised Figure 7 and Figure 7 – figure supplement 1.

**Recommendations for the authors:**

**Reviewer #1 (Recommendations for the authors):**
(1) Did you filter the neural signals during acquisition? If so, please include these details in the results.

Signals were bandpass-filtered (2.5 Hz to 7.6 KHz) at the hardware level at acquisition (with no additional software filtering applied), as now clarified in the Methods Electrophysiological recordings section as requested (page 12, lines: 525-526).

**Reviewer #2 (Recommendations for the authors):**
(1) Some studies have found evidence for excitatory projection neurons expressing PV and in particular some excitatory pyramidal cells can be labelled in PV-Cre mice. The authors might want to check if this is the case in their study, and if so, whether that might impact any conclusions.

Please see above for our response to this issue.

(2) I think the analysis shown in Figure S1 apparently reporting the absence of movements evoked by the forepaw stimulation could be strengthened. It is unclear what is shown in the various panels. I would imagine that an average of many stimulus repetitions would be needed to indicate whether there is an evoked movement or not. This could also be state-dependent and perhaps more likely to happen early in a recording session. Videography could also be helpful.

Please see above for our response to this issue.

(3) Some similar aspects of the evoked responses, including triphasic dynamics, have been reported in whisker S1 and M1, and the authors might want to cite Sreenivasan et al., 2016.

As noted above, we now cite this study.